# Membrane binding properties of the cytoskeletal protein bactofilin

**Ying Liu[1†], Rajani Karmakar[2], Maria Billini[1], Wieland Steinchen[3,4], Saumyak Mukherjee[2‡], Rogelio Hernandez-Tamayo[5,6], Thomas Heimerl[4], Gert Bange[3,4,7], Lars V Schäfer[2], Martin Thanbichler[1,4,6]\***

[1]Department of Biology, Marburg University, Marburg, Germany; [2]Theoretical Chemistry, Ruhr University Bochum, Bochum, Germany; [3]Department of Chemistry, Marburg University, Marburg, Germany; [4]Center for Synthetic Microbiology (SYNMIKRO), Marburg, Germany; [5]Microcosm Earth Center, Marburg, Germany; [6]Max Planck Fellow Group Bacterial Cell Biology, Max Planck Institute for Terrestrial Microbiology, Marburg, Germany; [7]Max Planck Fellow Group Molecular Physiology of Microbes, Max Planck Institute for Terrestrial Microbiology, Marburg, Germany

**\*For correspondence:**
thanbichler@uni-marburg.de

**Present address:** [†]Institute of Experimental Medicine, Universitätsklinikum Schleswig-Holstein, Kiel, Germany; [‡]Department of Theoretical Biophysics, Max Planck Institute of Biophysics, Frankfurt am Main, Germany

**Competing interest:** The authors declare that no competing interests exist.

## eLife Assessment

This **valuable** study advances our understanding of how bactofilin cytoskeletal proteins associate with cell membranes by identifying and characterizing a conserved membrane-targeting sequence. The evidence is **solid**, with a well-integrated combination of mutagenesis, biophysical analysis, molecular simulations, and bioinformatics supporting the mechanistic model. The work will be of particular interest to microbiologists and structural biologists studying bacterial cytoskeletons and membrane-protein interactions.

**Abstract** Bactofilins are a widespread family of cytoskeletal proteins that are essential for bacterial morphogenesis, chromosome organization, and motility. They assemble into non-polar filaments independently of nucleotides and typically associate with the cytoplasmic membrane. Their membrane interaction is thought to involve a short N-terminal peptide, but the underlying mechanism is unclear. Here, we clarify the complete membrane-targeting sequence (MTS) of the *Caulobacter crescentus* bactofilin BacA and identify residues critical for its function. Using molecular dynamics simulations, we show that its affinity for membranes arises from hydrophobic residue-driven water exclusion and electrostatic interactions with negatively charged phospholipid head-groups. Bioinformatic analysis suggests that this mode of membrane binding is conserved across diverse bacterial phyla. Importantly, we observe that BacA polymerization and membrane binding stimulate each other, and both of these processes are necessary for recruiting the membrane-bound client protein PbpC, a cell wall synthase that interacts with BacA via its N-terminal cytoplasmic region. PbpC can functionally replace the MTS of BacA when overproduced, demonstrating that client proteins contribute to the bactofilin-membrane association. Thus, bactofilin assembly and localization are determined by a complex interplay of different factors, thereby enabling the adaptation of these processes to the needs of the systems they control.

## Introduction

Cytoskeletal proteins mediate a variety of fundamental cellular processes in bacteria, including cell growth, cell division, and DNA segregation (*Cabeen and Jacobs-Wagner, 2010*; *Wagstaff and Löwe, 2018*). Many important components of the bacterial cytoskeleton have eukaryotic homologs. The

conserved cell division protein FtsZ, for instance, is a member of the tubulin superfamily (*Löwe and Amos, 1998*; *Mukherjee et al., 1993*), forming dynamic polymers at the cell division site that control the assembly and function of the cell division apparatus (*McQuillen and Xiao, 2020*). By contrast, the morphogenetic protein MreB, which governs cell growth and morphology in most rod-shaped bacteria (*Jones et al., 2001*; *van den Ent et al., 2001*), is homologous to actin (*Rohs and Bernhardt, 2021*). Some bacteria also possess intermediate-filament-like proteins, such as the coiled-coil-rich protein crescentin, which promotes cell curvature in the crescent-shaped alphaproteobacterium *Caulobacter crescentus* (*Ausmees et al., 2003*; *Liu et al., 2024*). Apart from these universally conserved cytoskeletal protein families, bacterial cells can contain various polymer-forming proteins that are typically absent from eukaryotes (*Wagstaff and Löwe, 2018*). A prominent member of this group of proteins is bactofilin (*Kühn et al., 2010*).

Bactofilin homologs are widespread among bacteria and involved in diverse cellular functions. In many species, they have important roles in cell shape determination, including the establishment or adjustment of cell curvature in *Helicobacter pylori* (*Sycuro et al., 2010*), *Campylobacter jejuni* (*Frirdich et al., 2023*), *Leptospira biplexa* (*Jackson et al., 2018*), and *Rhodospirillum rubrum* (*Pöhl et al., 2024*), the modulation of cell size in *Chlamydia trachomatis* (*Brockett et al., 2021*), the formation of buds in *Hyphomonas neptunium* (*Pöhl et al., 2024*), the stabilization of rod shape in *Myxococcus xanthus* (*Koch et al., 2011*) and *Proteus mirabilis* (*Hay et al., 1999*) and the synthesis of stalk-like cellular extensions in *C. crescentus* (*Kühn et al., 2010*), *Asticcacaulis biprosthecum* (*Caccamo et al., 2020*), and *Rhodomicrobium vannielii* (*Richter et al., 2023*). Other functions include DNA organization in *M. xanthus* (*Anand et al., 2020*; *Lin et al., 2017*) as well as cell motility in *Bacillus subtilis* (*El Andari et al., 2015*). Bactofilins are relatively small proteins characterized by a central Bactofilin A/B domain (InterPro ID: IPR007607) that is typically flanked by unstructured terminal regions of varying lengths. The central domain is ~110 amino acids long and folds into a compact right-handed β-helix with a triangular geometry, measuring roughly 3 nm along its longitudinal axis (*Kassem et al., 2016*; *Shi et al., 2015*; *Vasa et al., 2015*; *Zuckerman et al., 2015*). This basic unit polymerizes into extended non-polar filaments through alternating head-to-head and tail-to-tail interactions between individual subunits (*Deng et al., 2019*). The resulting protofilaments can further assemble into higher-order structures such as bundles, lattices, and 2D crystalline sheets in vitro (*Holtrup et al., 2019*; *Kühn et al., 2010*; *Pöhl et al., 2024*; *Sichel et al., 2022*; *Vasa et al., 2015*; *Zuckerman et al., 2015*) and, potentially, also in vivo (*Kühn et al., 2010*). Previous studies have indicated that bactofilins are typically associated with the cytoplasmic membrane (*Deng et al., 2019*; *Hay et al., 1999*; *Koch et al., 2011*; *Kühn et al., 2010*; *Lee et al., 2023*). However, only a small subset of them, including certain gammaproteobacterial representatives (*Hay et al., 1999*), contain predicted transmembrane helices, suggesting that canonical bactofilins use an unconventional membrane-targeting mechanism that may have specifically evolved in this cytoskeletal protein family. Consistent with this notion, the membrane-binding activity of bactofilin homologs from *Thermus thermophilus* (*Deng et al., 2019*) and *C. trachomatis* (*Lee et al., 2023*) has recently been shown to depend on a short peptide in the N-terminal unstructured region of the proteins. However, the conservation of these peptides, the precise mechanism underlying their affinity for the lipid bilayer, and the interplay between membrane attachment and bactofilin polymerization still remain to be investigated.

In this study, we comprehensively analyze the membrane-binding behavior of bactofilins, using BacA from *C. crescentus* as a model protein. Previous work has shown that BacA forms membrane-associated polymeric sheets at the old pole of the *C. crescentus* cell, which recruit the cell wall synthase PbpC to stimulate stalk formation at this subcellular location (*Billini et al., 2019*; *Hughes et al., 2013*; *Kühn et al., 2010*). By systematically exchanging residues in the N-terminal region of BacA, we identify a short N-terminal peptide (MFSKQAKS) that acts as the membrane-targeting sequence (MTS) of BacA and pinpoint residues critical for its membrane-binding activity in vivo and in vitro. We then clarify the molecular mechanism underlying the membrane affinity of this sequence using molecular dynamics simulations. A comprehensive bioinformatic analysis suggests that the mode of action identified for the MTS of BacA may be broadly conserved among bactofilins from various different phyla. Importantly, our findings indicate that membrane association and polymerization are cooperative processes that are both required for robust BacA assembly, with polymerization in turn promoting the interaction of BacA with its client protein PbpC, and vice versa. Together, these findings reveal the mechanistic basis of membrane binding by bactofilin homologs and highlight its implications for

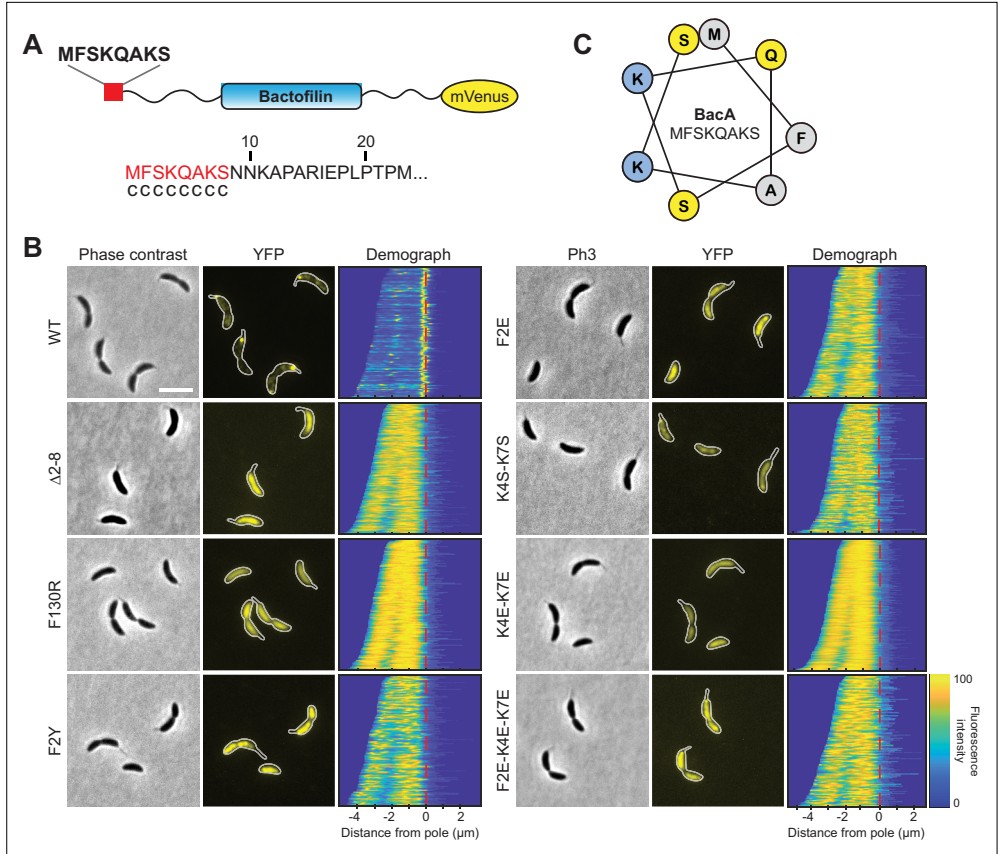

**Figure 1.** Identification of residues critical for the membrane-binding activity of BacA in vivo. (**A**) Schematic representation of the BacA-mVenus fusion protein used in this study. The proposed membrane-targeting sequence is highlighted in red. The sequence at the bottom shows that result of an amphipathic helix prediction for BacA using the AmphipaSeeK software (*Sapay et al., 2006*). Residues predicted to be located in an unstructured, randomly coiled region are labeled with 'c'. (**B**) Localization patterns of mutant BacA-mVenus variants. Δ*bacAB* cells producing BacA-mVenus or mutant variants thereof (strains LY84, LY89, LY90, LY97, LY111, LY112, LY113, LY119) were analyzed by phase contrast and fluorescence microscopy. The outlines of the cells are shown in the fluorescence images. Demographs summarizing the single-cell fluorescence profiles obtained from random subpopulations of cells are given next to the respective fluorescence images. The numbers of cells analyzed are: WT (130), Δ2–8 (292), F130R (156), F2Y (138), F2E (194), K4S-K7S (151), K4E-K7E (382), F2E-K4E-K7E (130). The vertical red line indicates the junction between the cell body and the stalk. Scale bar: 2 µm. (**C**) Helical wheel diagram of the first eight amino acids of BacA. Residues are colored by properties: hydrophobic (gray), basic (blue), uncharged (yellow).

The online version of this article includes the following source data and figure supplement(s) for figure 1:

**Figure supplement 1.** Ultrastructure of different BacA variants.

**Figure supplement 2.** Size-exclusion chromatography analysis of wild-type BacA and its F130R variant.

**Figure supplement 3.** Stability of different BacA-mVenus variants.

**Figure supplement 3—source data 1.** PDF file containing annotated original images of the western blots shown in *Figure 1—figure supplement 3*.

**Figure supplement 3—source data 2.** Original images of the western blots shown in *Figure 1—figure supplement 3*.

**Figure supplement 4.** Localization patterns of different BacA-mVenus variants.

bactofilin function, thereby shedding light on a thus-far poorly understood aspect in the biology of this widespread cytoskeletal protein.

## Results

### The membrane-binding activity of BacA depends on an N-terminal amphipathic peptide

Previous work has proposed a small N-terminal peptide as the membrane-targeting sequence (MTS) of canonical bactofilin homologs (*Deng et al., 2019*). In the case of *C. crescentus* BacA, the suggested region comprises the sequence MFSKQAKS at the N-terminal end of the protein (*Figure 1A*). To verify the importance of this peptide for BacA function, we compared the localization pattern of wild-type BacA with that of a mutant derivative lacking the predicted MTS. For this purpose, the two proteins were fused to the monomeric yellow fluorescent protein mVenus and produced in a strain carrying an in-frame deletion in the endogenous *bacA* and *bacB* genes (*Figure 1B*). As expected, the wild-type protein was fully functional and formed distinct fluorescent foci at the stalked pole of the cells, which have been previously shown to reflect the formation of small patches of membrane-associated BacA polymers (*Kühn et al., 2010*). The mutant protein (Δ2–8), by contrast, was evenly distributed in the cytoplasm, supporting the idea that the N-terminal region of BacA is involved in membrane binding. The diffuse localization also suggests a defect in polymerization, which, however, may be a secondary effect resulting from the loss of membrane association, since previous work has shown that the unstructured terminal regions are not essential for bactofilin assembly (*Sichel et al., 2022*; *Vasa et al., 2015*). In line with this finding, transmission electron microscopy (TEM) analysis confirmed that the Δ2–8 variant formed polymers similar to those of wild-type BacA when overproduced in *Escherichia coli* (*Figure 1—figure supplement 1*). Thus, although not essential, membrane binding may facilitate BacA polymerization at native expression levels by promoting its enrichment at the membrane surface and thereby stimulating inter-subunit interactions. To further evaluate the role of polymerization in BacA localization, we generated a mutant variant of BacA impaired in self-assembly by replacing a conserved phenylalanine residue in the C-terminal polymerization interface of its Bactofilin A/B domain (F130 in *C. crescentus* BacA) with a charged arginine residue. Consistent with previous work in other species (*Deng et al., 2019*; *Jacq et al., 2025*; *Zuckerman et al., 2015*), TEM analysis (*Figure 1—figure supplement 1*) and size-exclusion chromatography studies (*Figure 1—figure supplement 2*) verified that the mutant protein (F130R) failed to form polymeric structures in vitro. Importantly, the corresponding mVenus fusion protein no longer formed membrane-associated polar foci but was dispersed within the cytoplasm (*Figure 1B*), suggesting that polymerization and membrane binding could stimulate each other.

Apart from transmembrane segments, proteins commonly employ amphiphilic helices to associate with the inner face of the cytoplasmic membrane, as exemplified by the bacterial actin homologs FtsA (*Pichoff and Lutkenhaus, 2005*) and MreB (*Salje et al., 2011*). As a first step to clarify the mechanism underlying the membrane-binding activity of the MTS, we, therefore, determined whether it was able to adopt an amphiphilic helical structure. However, neither machine-learning-based pattern recognition, as employed by AmphipaSeeK (*Sapay et al., 2006*; *Figure 1A*), nor helical wheel analysis (*Figure 1C*) provided any support for this possibility, suggesting a different mode of action.

To obtain more insight into the function of the MTS, we systematically exchanged single or multiple amino acid residues in the N-terminal eight amino acids of BacA-mVenus (*Figure 1—figure supplement 3*) and analyzed the localization patterns of the mutant proteins. Among the single exchanges, substitution of the hydrophobic phenylalanine residue at position 2 (F2Y and F2E) led to a disperse or, occasionally, patchy distribution of the fusion protein (*Figure 1B*). Other single substitutions, by contrast, had relatively mild effects, including the formation of multiple and/or mispositioned bactofilin complexes (*Figure 1—figure supplement 4*). Notably, however, a double exchange replacing the two positively charged lysine residues in the MTS with polar serine residues (K4S-K7S) strongly increased the fraction of delocalized protein (*Figure 1B*), although distinct fluorescent foci or patches were still detectable, suggesting residual membrane-binding activity. An even stronger effect, as reflected by a complete dispersion of the fluorescence signal, was observed when negatively charged glutamate residues were introduced at these positions (K4E-K7E). Collectively, these results point to a central role of F2 and K4/K7 in the association of BacA with the cytoplasmic membrane.

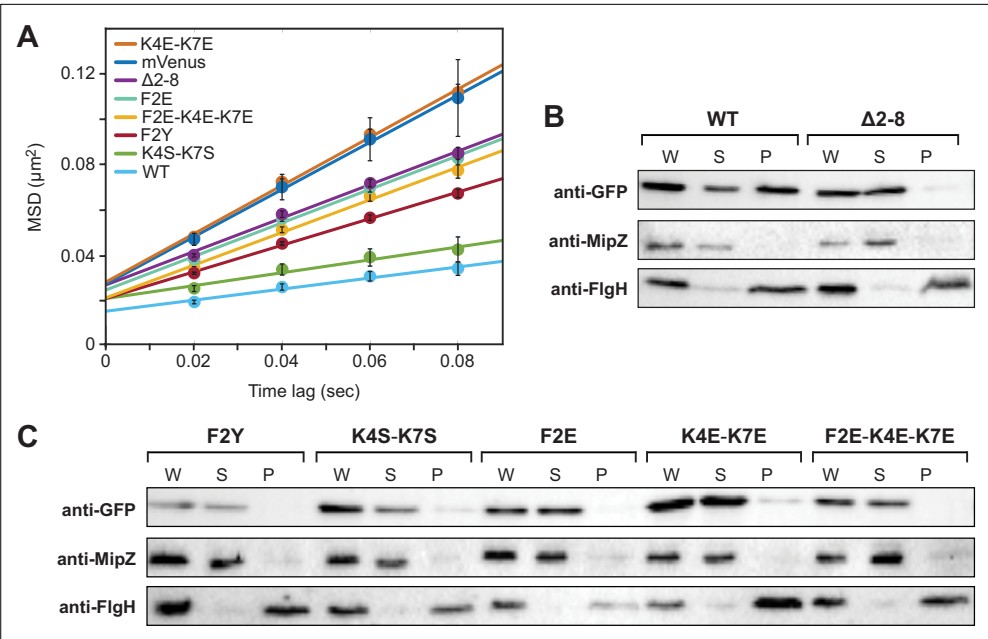

**Figure 2.** Verification of residues F2 and K4/K7 as critical components of the BacA membrane-targeting sequence (MTS). (**A**) Mobility of the indicated BacA-mVenus fusion proteins. Shown is the average mean-squared displacement (MSD) (± SD) as a function of time, based on single-particle tracking analysis. The fitted lines were obtained by linear regression analysis. (**B**) Cell fractionation experiment investigating the membrane-binding activity of BacA-mVenus (WT) or a mutant variant lacking the predicted MTS (Δ2–8). Whole-cell lysates (W) as well as the soluble (S) and pellet (P) fractions of cells producing the indicated proteins were subjected to immunoblot analysis with an anti-GFP antibody, detecting the BacA-mVenus fusion protein. As controls, the same samples were probed with antibodies raised against the soluble cell division regulator MipZ (***Thanbichler and Shapiro, 2006***) or the membrane-bound flagellar L-ring subunit FlgH (***Mohr et al., 1996***) from *C. crescentus*. (**C**) As in panel **B**, but for cells producing mutant BacA-mVenus variants with single or multiple amino-acid exchanges in the predicted MTS. Shown are representative images (n=3 independent replicates). The strains used are given in the legend to *Figure 1B*.

The online version of this article includes the following video, source data, and figure supplement(s) for figure 2:

**Source data 1.** PDF files containing annotated original images of the western blots shown in *Figure 2B and C*.

**Source data 2.** Original images of the western blots shown in *Figure 2B and C*.

**Figure supplement 1.** Subcellular localization of the single-molecule tracks obtained for different BacA-mVenus variants.

**Figure 2—video 1.** Single-particle dynamics of different BacA-mVenus variants.
https://elifesciences.org/articles/100749/figures#fig2video1

To further characterize the mutant BacA-mVenus variants, we analyzed their mobility within the cell using single-particle tracking (*Figure 2—video 1* and *Figure 2—figure supplement 1*). As expected, given its ability to interact with the membrane and polymerize, the wild-type protein exhibited a very low diffusional mobility. Mutant variants that were partially delocalized (F2Y and K4S-K7S) were significantly more mobile, and variants with a largely diffuse localization (Δ2–8, F2E, K4E-K7E, and F2E-K4E-K7E) displayed the highest diffusion rates, approaching the mobility of free mVenus (*Figure 2A* and *Supplementary file 1*). These results support the notion that the MTS is required for the formation of stable membrane-associated BacA assemblies.

As a direct means to analyze the membrane-binding activity of the fusion proteins, we next performed cell fractionation studies. To this end, cells producing different BacA-mVenus variants were lysed and membranes were separated from soluble components by ultracentrifugation. Western blot analysis revealed that a large part of the wild-type protein was detected in the membrane fraction, whereas the variant lacking the MTS (Δ2–8) was completely soluble (*Figure 2B*). A severe reduction in membrane-binding activity was also observed for all fusion proteins lacking residues F2 and K4/K7 (*Figure 2C*), confirming the importance of these residues in the function of the MTS.

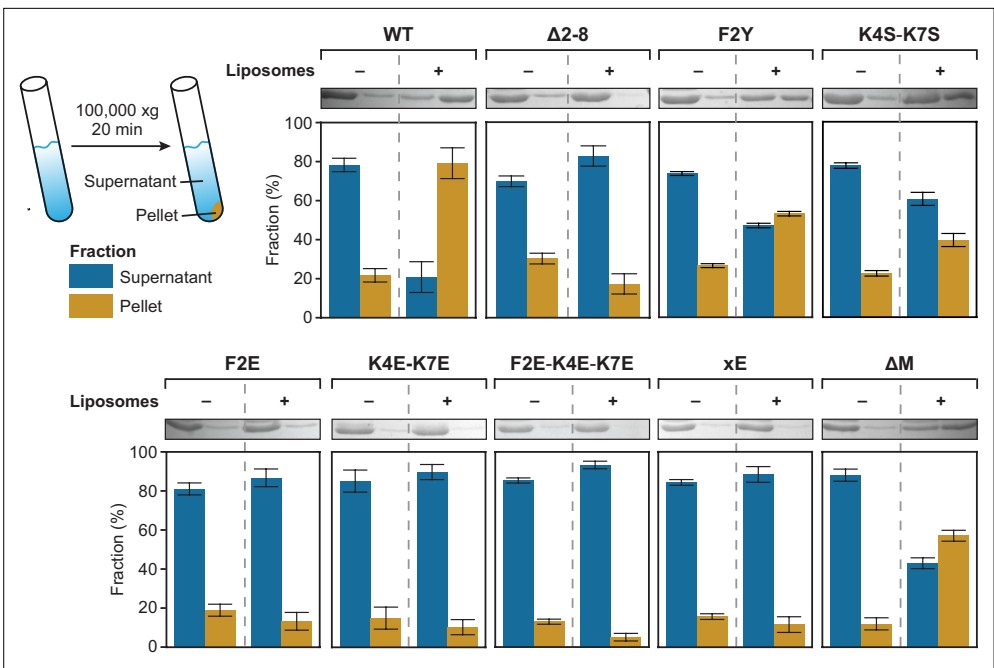

**Figure 3.** Co-sedimentation analysis of the association of various BacA variants with liposomes. The indicated proteins (20 µM) were incubated without (-) or with (+) liposomes (0.4 mg/mL) prior to ultracentrifugation. The supernatant and pellet fractions of each mixture were analyzed by SDS gel electrophoresis. Shown are scans of representative gels and a quantification of the average relative signal intensities (± SD) obtained for the different fractions (n=3 independent replicates).

The online version of this article includes the following source data for figure 3:

**Source data 1.** PDF file containing annotated original images of the SDS polyacrylamide gels shown in *Figure 3*.

**Source data 2.** Original images of the SDS polyacrylamide gels shown in *Figure 3*.

After having identified a role of the MTS in the association of BacA with membranes in vivo, we aimed to verify the membrane-targeting activity of this peptide in a defined system in vitro. For this purpose, we purified wild-type BacA and various mutant derivatives, using a cleavable N-terminal His$_6$-SUMO affinity tag (*Marblestone et al., 2006*), which allowed us to obtain proteins without any non-native extensions that could potentially interfere with the analysis. This approach also made it possible to obtain two additional BacA variants, one lacking the N-terminal methionine of BacA (ΔM) and another one lacking the N-terminal methionine and containing a glutamate instead of the phenyl-alanine normally located at the second position of the MTS (xE). We then incubated the different proteins with small unilamellar vesicles (liposomes) made of phosphatidylglycerol, which has been identified as the most abundant membrane lipid in *C. crescentus* under standard growth conditions (*Contreras et al., 1978*; *Stankeviciute et al., 2019*). Subsequently, the liposomes were collected by ultracentrifugation and analyzed for the amount of bound protein (*Figure 3*). Consistent with the in vivo data, a large proportion (~80%) of the wild-type protein was associated with the liposome pellet, whereas only a small fraction sedimented in control reactions lacking liposomes, likely reflecting large, poorly soluble polymer bundles. Variants lacking the entire MTS (Δ2–8) or carrying the F2E or K4E-K7E exchanges, by contrast, showed severe defects in membrane binding, with sedimentation efficiencies similar to those obtained in the liposome-free control reactions. Residual binding was still observed for the F2Y and K4S-K7S variants, consistent with the milder defects observed for these proteins in the localization and single-particle tracking studies (compare *Figures 1B and 2A*). Notably, while the behavior of the xE variant closely resembled that of the F2E variant, the absence of residue M1 (ΔM) alone only partially abolished the membrane association of BacA, indicating that the N-terminal methi-onine plays an important but not decisive role in the function of the MTS (*Figure 3*). Together, these findings verify the existence of an N-terminal MTS in BacA. Moreover, they identify the hydrophobic

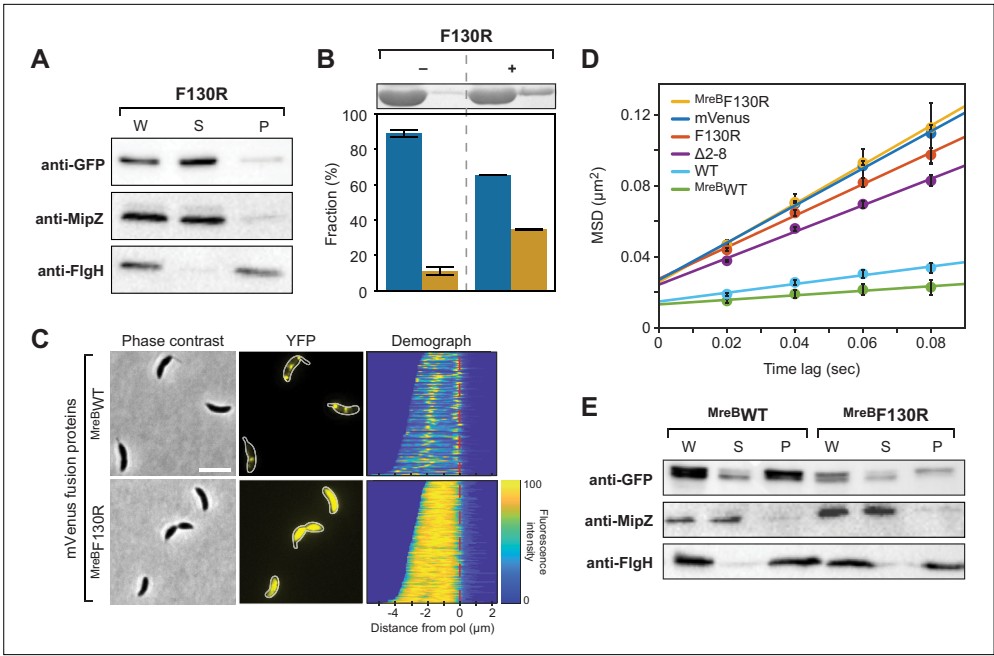

**Figure 4.** Interplay between BacA assembly and membrane binding. (**A**) Cell fractionation experiment investigating the membrane-binding activity of the polymerization-deficient F130R variant of BacA-mVenus in vivo (LY119). The analysis was performed as described for *Figure 2B*. (**B**) Co-sedimentation analysis of the association of BacA-F130R with liposomes in vitro, performed as described for *Figure 3*. (**C**) Role of polymerization in the membrane association of a BacA-mVenus variant carrying the membrane-targeting sequence of *E. coli* MreB. Shown are phase contrast and fluorescence images of ΔbacAB mutants (LY103, LY123) producing either a BacA-mVenus variant in which the MTS is replaced by two tandem copies of the N-terminal amphiphilic helix of *E. coli* MreB (MreBWT) or a polymerization-deficient variant thereof (MreBF130R). Demographs summarizing the single-cell fluorescence profiles obtained from random subpopulations of cells are given next to the respective fluorescence images. The number of cells analyzed are: MreBWT (126), MreBF130R (169). The vertical red line indicates the junction between the cell body and the stalk. Scale bar: 2 µm. (**D**) Mobility of the indicated BacA-mVenus fusion proteins. Shown is the average mean-squared displacement (MSD) (± SD) as a function of time, based on single-particle tracking analysis. The mVenus, Δ2–8, and WT data are taken from *Figure 2A* and shown for comparison. (**E**) Cell fractionation experiment investigating the membrane-binding activity of MreBBacA-mVenus (MreBWT) and its polymerization-deficient F130R variant (MreBF130R) in vivo (LY103, LY123). The analysis was performed as described for *Figure 2B*.

The online version of this article includes the following source data and figure supplement(s) for figure 4:

**Source data 1.** PDF files containing annotated original images of the western blots shown in *Figure 4A and E*.

**Source data 2.** Original images of the western blots shown in *Figure 4A and E*.

**Figure supplement 1.** Solubility of different CreS-mNeonGreen variants.

**Figure supplement 1—source data 1.** PDF file containing annotated original images of the western blots shown in *Figure 4—figure supplement 1B*.

**Figure supplement 1—source data 2.** Original images of the western blots shown in *Figure 4—figure supplement 1B*.

residues M1 and F2 and the two positively charged lysine residues K4 and K7 as key determinants of its membrane-binding affinity.

## BacA polymerization and membrane binding are mutually stimulating processes

Our results showed that the disruption of the polymerization interface (F130R) resulted in a diffuse, cytoplasmic localization of BacA, suggesting that polymerization might be a prerequisite for efficient membrane binding (*Figure 1B*). To investigate this possibility, we clarified the membrane-binding activity of the polymerization-deficient F130R variant of BacA-mVenus. Cell fractionation experiments

showed that the fusion protein was barely detectable in the membrane fraction (*Figure 4A*). In line with this finding, only marginal membrane-binding activity was observed for the untagged BacA-F130 variant in liposome-binding assays (*Figure 4B*). These results suggest that the MTS has a relatively low affinity for the membrane, so that a stable interaction can only be achieved by cooperative binding of multiple MTS-containing N-terminal tails, arrayed on the surface of bactofilin polymers. Further support for this idea came from domain swapping experiments that showed that the MTS of BacA was able to mediate stable membrane attachment when fused to a derivative of the polymer-forming protein crescentin from *C. crescentus* lacking its native membrane-targeting sequence (*Figure 4—figure supplement 1*).

Interestingly, the diffuse localization and high diffusional mobility of BacA variants containing a defective MTS (*Figures 1B and 2A*) indicate that there may also be a converse stimulatory effect of membrane binding on BacA polymerization. To further assess this possibility, we analyzed whether an MTS-defective BacA variant could regain the ability to form membrane-associated polymeric complexes if equipped with a heterologous membrane-targeting sequence. To this end, two copies of the N-terminal amphiphilic helix of *E. coli* MreB (*Salje et al., 2011*) were fused to a BacA-mVenus variant lacking the native MTS-containing peptide (Δ2–8). Microscopic analysis revealed that the fusion protein condensed into distinct foci at the old pole or the cell center (*Figure 4C*), reminiscent of the aberrant localization patterns observed for mutant proteins with exchanges in the MTS that did not completely abolish membrane binding (compare *Figure 1—figure supplement 4*). The restoration of fluorescent foci was accompanied by a strong decrease in diffusional mobility, as determined by single-molecule tracking (*Figure 4D*). In addition, cell fractionation experiments showed that the fusion protein was highly enriched in the membrane pellet (*Figure 4E*). Importantly, when its polymerization interface was disrupted by introduction of the F130R exchange, focus formation was abolished, membrane binding was strongly reduced, and the diffusional mobility of the fusion protein increased to a value similar to that of an F130R variant containing the native MTS of BacA (*Figure 4C–E*). Collectively, these findings demonstrate that the polymerization and membrane-binding activities of BacA strongly stimulate each other.

## Molecular dynamics simulations identify M1, K4, and K7 as key residues of the BacA MTS

To investigate the molecular underpinnings of the interaction between the MTS of BacA and the membrane, we turned to all-atom molecular dynamics (MD) simulations. For this purpose, peptides comprising the ten N-terminal bactofilin residues were simulated in the presence of a multi-component lipid bilayer that was composed of approximately 50% monoglucosyldiglyceride (GLY), 33% phosphatidylglycerol (PG), and 16% diacylglycerol (DAG) lipids (see *Supplementary file 2* for details), approximating the native composition of *C. crescentus* cell membranes under standard growth conditions (*Chow and Schmidt, 1974*; *Contreras et al., 1978*; *Stankeviciute et al., 2019*). In addition to the wild-type peptide MFSKQAKSNN, our analysis also included the mutant F2Y and K4S-K7S variants (each in a separate simulation).

The starting structures of the peptides were modeled in an extended conformation without any secondary structure. At the beginning, a single peptide (wild-type, F2Y, or K4S-K7S) was placed in the aqueous phase at a distance of approximately 3 nm from the lipid headgroups. During the 500-ns simulations, all three peptides rapidly bound to the membrane and established multiple contacts with the lipids at the membrane/water interface (illustrated for the wild-type MTS in *Figure 5A*). In doing so, the peptides did not adopt any persistent secondary structures but remained disordered, with a preference for somewhat extended conformations (*Figure 5B and C* and *Supplementary file 3*). Notably, in control simulations in which the peptides were initially modeled as α-helices, the peptides unfolded rapidly, either in the aqueous phase or after binding to the membrane/water interface, eventually yielding the same results (within the statistical uncertainties) as the simulations starting from an extended conformation.

Excluding the initial phase of the simulations in which the peptides were not yet bound to the membrane, we then determined the density profiles of the three different peptides along the membrane normal (*Figure 5D* and *Figure 5—figure supplement 1*). The results show that the N-terminal residues 1 and 2 insert deeply into the lipid bilayer, assuming positions below the phosphate headgroup region, while the positions of the remaining residues (3–10) are gradually shifted towards

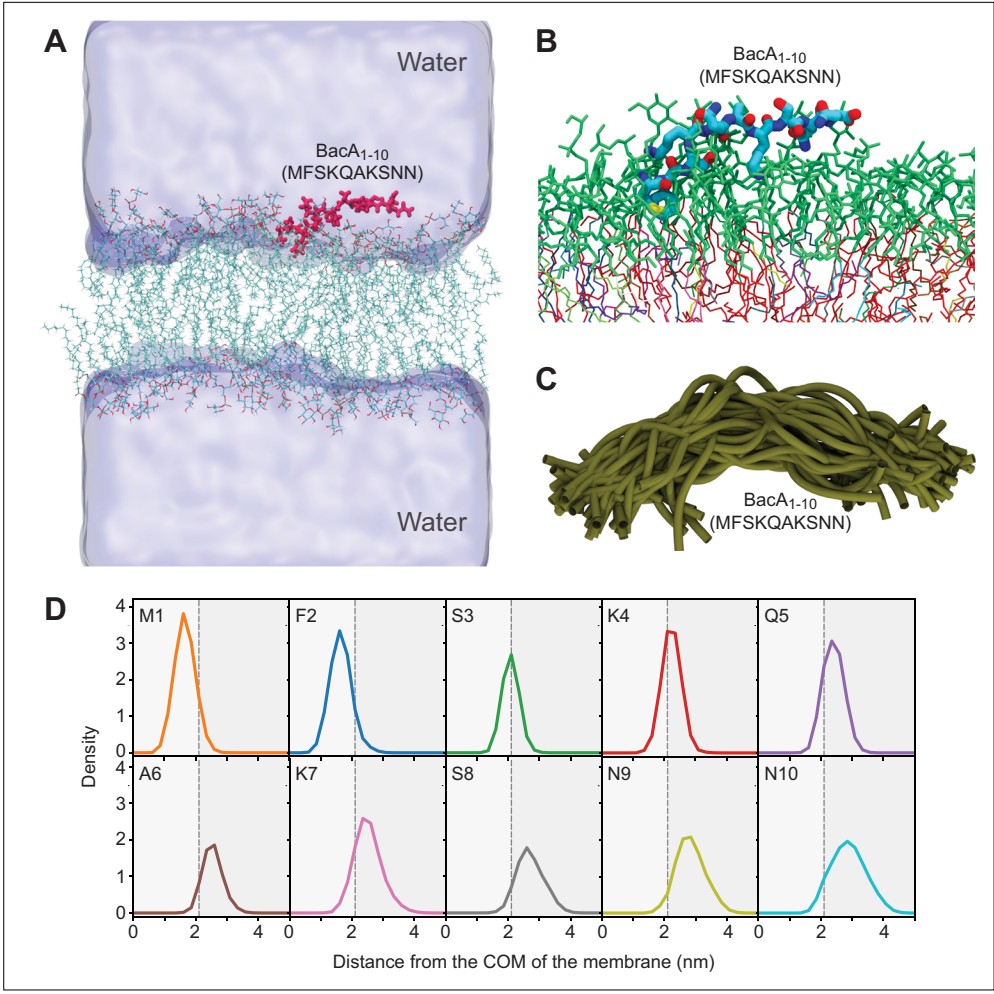

**Figure 5.** Molecular dynamics simulation of the interaction between the BacA membrane-targeting sequence (MTS) and a model membrane. (**A**) Snapshot of the molecular dynamics (MD) simulation system showing the 10-mer peptide MFSKQAKSNN (BacA$_{1-10}$; red) after binding to the lipid bilayer. The water is shown in surface representation. K$^+$ and Cl$^-$ counterions are not shown. (**B**) Close-up view of *a representative snapshot from the MD simulation visualizing the binding mode of the peptide on the membrane surface.* (**C**) *Structural overlay of 40 snapshots from the MD simulation, taken after constant time intervals from the trajectory.* (**D**) *Density profiles of individual residues in the wild-type peptide along the membrane normal, i.e., the z-component of the distance vector from the center-of-mass (COM) of the bilayer, with the membrane midplane located at zero. The vertical dashed black line indicates the maximum of the density distribution of the lipid headgroup phosphates.*

The online version of this article includes the following figure supplement(s) for figure 5:

**Figure supplement 1.** Density profiles for different membrane-targeting sequence (MTS) variants determined by molecular dynamics simulation.

the aqueous phase. The peptides thus bind to the membrane in a tilted orientation, with their N-terminal region protruding more deeply into the hydrophobic parts of the lipid headgroup region (*Figure 5B*). Interestingly, in case of the F2Y variant, the presence of tyrosine at position 2 pulls residue M1 and Y2 further towards the headgroup/water interface (*Figure 5—figure supplement 1B*). This result can be explained by the preferential hydration of tyrosine compared to phenylalanine, as is also reflected in the difference of the solvation free energies ($\Delta G_{solv}$) of the corresponding side-chain analogs *p*-cresol (–25.6 kJ mol$^{-1}$) and toluene (–3.2 kJ mol$^{-1}$) (*Wolfenden et al., 1981*).

To characterize the interactions between the peptides and the membrane in more detail, the contacts between the individual peptide residues and the different lipid species in the bilayers were counted. The results confirm the above finding that residues 1 and 2 show the strongest association with lipids and that lipid interactions gradually decrease for residues further down the peptide

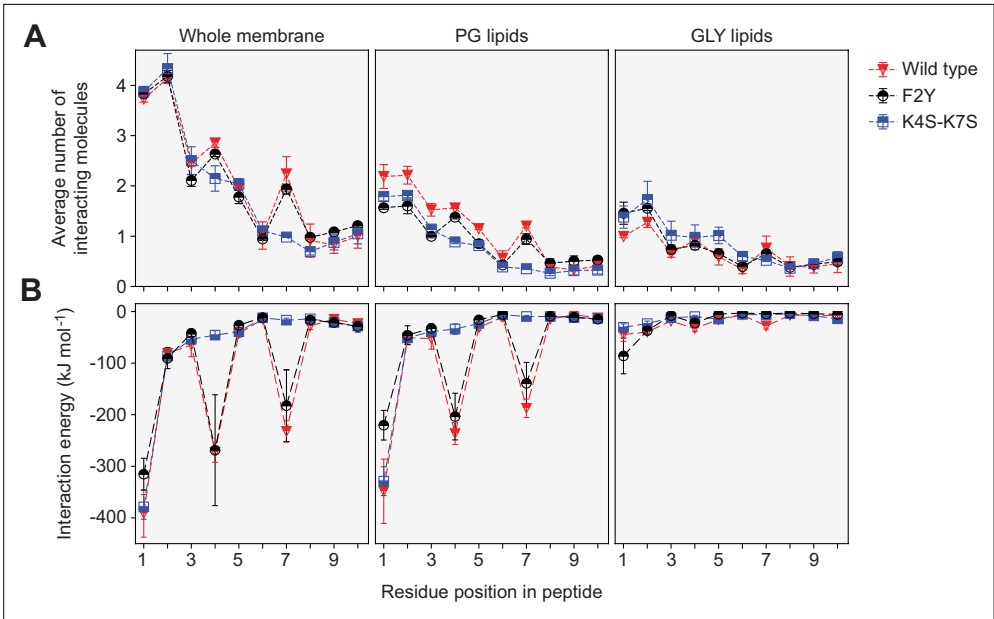

**Figure 6.** Contact numbers and interaction energies for different peptide-lipid bilayer interactions. (**A**) The graph shows the total number of contacts between individual residues in the wild-type, K4S-K7S, and F2Y peptides and the lipid bilayer, as well as the number of contacts with phosphatidylglycerol (PG) lipids and monoglucosyldiglyceride (GLY) lipids. A contact between a peptide residue and a lipid was defined to exist if any two non-hydrogen atoms of the residue and a lipid molecule were within a distance of 0.5 nm to each other. Contacts were counted for each frame of the MD trajectories and averaged. Multiple contacts between a peptide and a lipid molecule were treated as a single contact, so that the number of contacts counted was either 1 or 0. The statistical errors plotted were obtained from the difference between the two different sets of 500-ns simulations, starting with peptides in an unfolded or α-helical conformation, respectively. (**B**) Energies of the interactions between individual residues in the wild-type, K4S-K7S, and F2Y peptides and the lipid bilayer. The interaction energies plotted are the combined interaction energies of all Coulomb and van der Waals interactions in the force field averaged over the simulation trajectories.

chain (*Figure 6A*). Notably, among these interactions, contacts with anionic PG lipids dominate, even though the percentage of PG lipids (33%) in the simulated bilayer is smaller than that of GLY lipids (50%) (see above and *Supplementary file 2*), which can largely be attributed to the lysine residues at positions 4 and 7. In line with this result, the K4S-K7S variant shows a strong reduction in the number of PG contacts at these positions, which is partially compensated by a higher prevalence of GLY contacts, particularly for residues 1–5 (*Figure 6A*). The F2Y variant, by contrast, did not show a marked decrease in the number of total lipid contacts for position 2, with fewer PG contacts (of residues 1 and 2) again compensated by more GLY lipid contacts (*Figure 6A*). Taken together, the lipid contact analysis shows that the N-terminal residues M1 and F2 establish the closest contacts with the lipid bilayer, and that the lysine residues K4 and K7 strongly interact with PG lipids.

The contact analysis described above only provides insight into the spatial proximity of the peptides and the lipids. To additionally determine the strength of the interactions, we analyzed the MD simulations in terms of the interaction energies between the different peptide residues and the lipid molecules (*Figure 6B*). For both the wild-type peptide and its F2Y variant, the strongest (most favorable) peptide-lipid interaction energy is seen for residues 1, 4, and 7, which can be attributed to strong electrostatic attraction between the PG lipids and the charged $NH_3^+$ group at the N-terminus (residue M1) as well as the side chains of residues K4 and K7, respectively. Notably, out of all residues, M1 interacts most strongly with the membrane. Nevertheless, its removal (ΔM) had only a moderate effect on the membrane-binding activity of BacA in vitro (*Figure 3*), corroborating the notion that the largest part of the interaction energy is provided by the N-terminal $NH_3^+$ group of M1 rather than its hydrophobic side chain. However, taken together, the two lysines K4 and K7 make an even larger contribution to the interaction, consistent with the significant reduction in membrane association observed for the K4S-K7S variants of BacA (*Figure 1B*, *Figure 2C* and *Figure 3*). It is interesting

to note that in terms of the interaction energy, residue F2 does not seem to play a prominent role in the peptide-membrane association (*Figure 6B*), although the F2Y exchange strongly reduces the membrane-binding affinity of BacA (*Figure 1B*, *Figure 2C* and *Figure 3*). The deleterious effect of this exchange may be explained by the preferential hydration of tyrosine compared to phenylalanine, which is not reflected in the peptide-membrane interaction energy but in the difference in hydration free-energy (see above). Furthermore, residues M1 and Y2 in the F2Y peptide are located in closer proximity to the aqueous phase (see *Figure 5—figure supplement 1B*), which leads to a substantial decrease in the favorable interaction energy of the N-terminal $NH_3^+$-group with PG lipids that is only incompletely compensated by slightly more favorable interactions with GLY lipids (*Figure 6B*). Together, these results provide a rational basis for the experimental finding that the F2Y and K4S-K7S variants of BacA have a considerably lower membrane-binding affinity than the wild-type protein.

## The N-terminal membrane-targeting sequence is conserved among bactofilin homologs

The results of the MD simulations confirmed that the short peptide (MFSKQAKS) at the N-terminus of BacA acts as an MTS. To assess whether this mode of interaction was more widespread among bactofilins, we established a bioinformatic pipeline to analyze the conservation of the N-terminal region of bactofilin homologs from a broad range of bacterial lineages (*Figure 7—figure supplement 1*). To this end, all proteins containing an annotated bactofilin domain were retrieved from the UniProt database (*Bateman et al., 2023*). After the exclusion of entries from non-bacterial origin, we eliminated all bactofilin homologs containing predicted N-terminal transmembrane helices as a membrane anchor. These proteins were relatively rare (~3.2%) but widely distributed across the bacterial phylogeny, with the majority found in the gammaproteobacteria (*Figure 7—figure supplement 1*). To avoid biases arising from the over-representation of certain bacterial species in UniProt, we next grouped highly similar (>90% identity) bactofilin sequences, which typically represent orthologs from different sequenced strains, and only kept one sequence per group. For each of the remaining proteins, we then determined the N-terminal region preceding the bactofilin domain. Based on the length of the

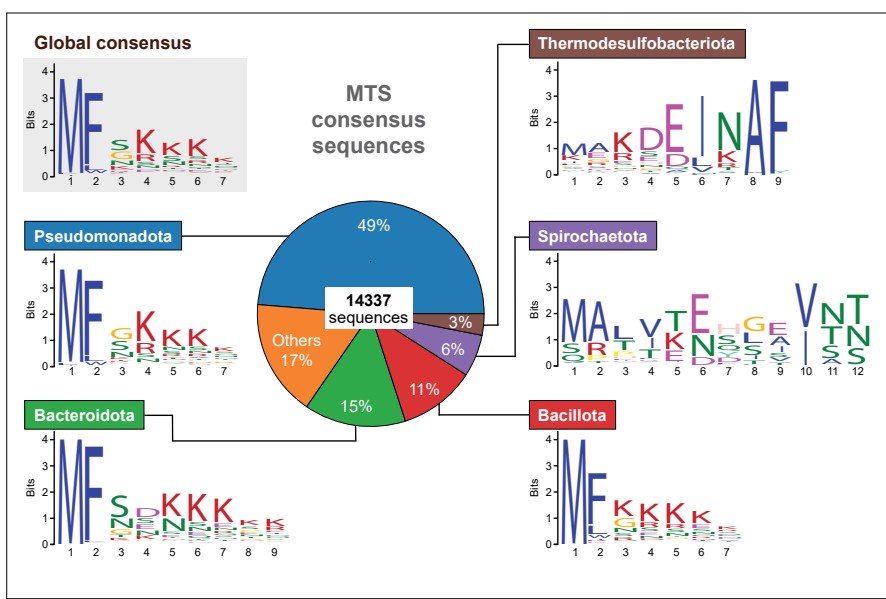

**Figure 7.** Conservation of the N-terminal regions of bactofilin homologs in different bacterial phyla. The pie chart in the middle shows the relative distribution of the 14,337 unique bactofilin homologs analyzed among the indicated bacterial phyla. The sequence logos give the most widespread N-terminal motifs obtained either by a global analysis of all 14,337 bactofilin sequences (global consensus) or by an analysis of subsets of these sequences from specific phyla.

The online version of this article includes the following figure supplement(s) for figure 7:

**Figure supplement 1.** Assessment of the conservation of the bactofilin membrane-targeting sequence.

**Figure supplement 2.** Conserved N-terminal motifs in bactofilin homologs from different phyla.

MTS identified for BacA, we eliminated proteins whose predicted N-terminal tails were shorter than eight amino acids, potentially due to misannotation of the translational start sites. The remaining proteins (14,337 sequences) were then used to search for conserved N-terminal amino acid motifs.

A global analysis of the entire protein set yielded a conserved motif with the consensus sequence MFSKKKK, which resembled the MTS identified for *C. crescentus* BacA and contained both the two hydrophobic N-terminal residues that insert into the core of the lipid bilayer as well as the two lysine residues that make electrostatic interactions with the headgroups of PG lipids (*Figure 7*). This result suggests that the presence of an N-terminal MTS is a common feature of bacterial bactofilin homologs. However, this global approach might obscure lineage-specific adaptations of the MTS to evolutionary differences in membrane lipid composition. To address this issue, we conducted an additional analysis in which we sorted the proteins according to their phylogenetic origin before performing a motif search. In doing so, we focused on the phyla Pseudomonadota, Bacteroidota, Bacillota, Spirochaetota, and Thermodesulfobacteriota, which contribute the majority of the bactofilin homologs in the UniProt database. This refined analysis confirmed the presence of conserved N-terminal motifs in each of the five phyla, although there is variability in the consensus sequences of these motifs both within and between phyla (*Figure 7—figure supplement 1*). A comparison of the most prevalent motifs (*Figure 7*) revealed that bactofilin homologs from Pseudomonadota and Bacillota share highly similar N-terminal sequences, which largely correspond to the global motif described above. A similar N-terminal motif is also found in most bactofilin homologs from Bacteroidota, although these proteins often display a negatively rather than a positively charged residue at position 4. The most prevalent motif identified in Spirochaetota and Thermodesulfobacteriota, by contrast, is fundamentally different from the global consensus motif and may thus have a function unrelated to membrane binding (*Figure 7*). However, there is also a subset of sequences from these phyla that do share MTS-like sequences (*Figure 7—figure supplement 2*), suggesting that the ability of bactofilins to interact with membranes is still widespread in these bacterial lineages. Together, these results strongly suggest that the majority of bactofilin homologs known to date feature an N-terminal MTS whose composition and mode of action are similar to that of *C. crescentus* BacA.

## BacA polymerization promotes client-protein binding

The only client protein of BacA reported to date is the cell wall synthase PbpC, a bitopic membrane protein related to *E. coli* PBP1A (*Yakhnina and Gitai, 2013*; *Strobel et al., 2014*). Previous work has shown that its N-terminal cytoplasmic tail was sufficient for its bactofilin-dependent recruitment to the stalked cell pole of *C. crescentus*, suggesting that this region of the protein contains all critical interaction determinants (*Hughes et al., 2013*; *Kühn et al., 2010*). However, its precise mode of interaction with the polar bactofilin cluster and the role of this interaction in bactofilin assembly has remained unclear.

To clarify which of the two bactofilin paralogs of *C. crescentus* interact with PbpC to mediate its polar localization, we analyzed the subcellular distribution of an mVenus-tagged PbpC variant in different bactofilin mutants. The fusion protein retained its wild-type localization pattern in the Δ*bacB* background but failed to condense into polar foci in Δ*bacA* cells, identifying BacA as its main interactor (*Figure 8* and *Figure 8—figure supplement 1A*). Next, we aimed to pinpoint the regions in the cytoplasmic tail of PbpC that mediate its recruitment to BacA. For this purpose, we retrieved the amino acid sequences of PbpC homologs and aligned their predicted cytoplasmic tails (*Figure 8—figure supplement 2*). This analysis identified four distinct segments, including (i) a highly conserved peptide at the N-terminal end of PbpC (region C1), (ii) a proline-rich medial region, (iii) a second highly conserved peptide (Region C2), and (iv) a region rich in positively charged amino acids immediately adjacent to the transmembrane helix. To assess the importance of these segments, we generated mVenus-PbpC derivatives whose cytoplasmic tail either lacked region C1 (Δ2–13) or only comprised region C1, with the remaining parts being replaced by an unstructured region taken from the periplasmic protein DipM of *C. crescentus* (*Izquierdo-Martinez et al., 2023*; *Figure 8—figure supplement 3*). Microscopic analysis showed that the removal of region C1 resulted in the loss of polar localization, whereas the chimeric variant showed the same localization pattern as the wild-type fusion protein (*Figure 8A*). Similar results were obtained for cells that were cultivated under phosphate-limiting conditions to stimulate stalk growth (*Schmidt, 1968*). Interestingly, the delocalization of mVenus-PbpC upon deletion of region C1 did not reduce its ability to promote stalk elongation

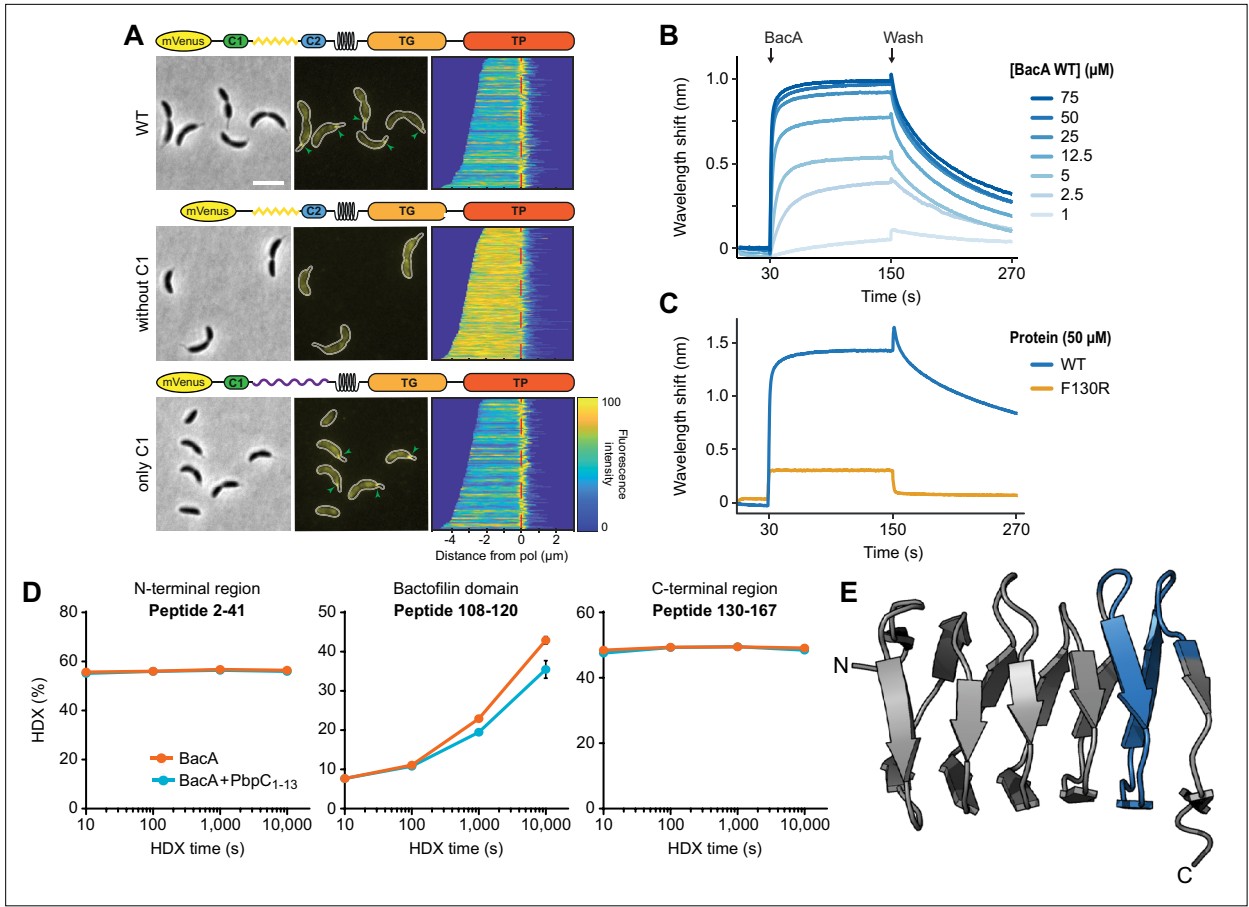

**Figure 8.** Interaction of BacA with its client protein PbpC. (**A**) Localization patterns of different PbpC variants. Δ*bacB* Δ*pbpC* cells producing mVenus-PbpC (LY75) or mutant variants thereof lacking region C1 (LY76) or carrying an unstructured region from *C. crescentus* DipM in place of the unstructured region connecting region C1 and the transmembrane helix (LY77) were analyzed by phase contrast and fluorescence microscopy. The outlines of the cells are shown in the fluorescence images. Demographs summarizing the single-cell fluorescence profiles obtained from random subpopulations of cells are given next to the respective fluorescence images. The number of cells analyzed are: LY75 (158), LY76 (253), LY77 (119). Scale bar: 2 µm. (**B**) Biolayer interferometric analysis of the interaction between PbpC$_{1-13}$ and BacA. A synthetic peptide comprising the first 13 amino acids of PbpC (PbpC$_{1-13}$) was immobilized on a biosensor and probed with increasing concentrations of BacA. After the association step, the sensor was transferred to a protein-free buffer to monitor the dissociation reaction. The graph shows a representative titration series (n=3 independent replicates). (**C**) Comparison of the interaction of PbpC$_{1-13}$ with BacA and its polymerization-deficient F130R variant, performed as described in panel B. (**D**) Mapping of the PbpC binding site on BacA by hydrogen-deuterium exchange (HDX) mass spectrometry. The plots show the extent of deuterium uptake by three representative peptides obtained after peptic digestion of BacA protein (2.5 µM) that had been incubated in the absence or presence of the PbpC$_{1-13}$ peptide (10 µM) for the indicated time periods (see *Supplementary file 4* for the full set of peptides). (**E**) Mapping of the differences in deuterium uptake observed at t=1000 s onto the solid-state NMR structure of BacA (*Shi et al., 2015*).

The online version of this article includes the following source data and figure supplement(s) for figure 8:

**Figure supplement 1.** Localization patterns of mVenus-PbpC and BacA-Venus in different strain backgrounds.

**Figure supplement 2.** Sequence alignment of the cytoplasmic tail of PbpC homologs.

**Figure supplement 3.** Stability of the mVenus-PbpC fusion proteins used in this study.

**Figure supplement 3—source data 1.** PDF file containing an annotated original image of the Western blot shown in *Figure 8—figure supplement 3*.

**Figure supplement 3—source data 2.** Original image of the Western blot shown in *Figure 8—figure supplement 3*.

**Figure supplement 4.** Relevance of BacA binding for the localization and functionality of PbpC under phosphate-limiting conditions.

**Figure supplement 4—source data 1.** PDF file containing an annotated original image of the Western blot shown in *Figure 8—figure supplement 4C*.

**Figure supplement 4—source data 2.** Original image of the Western blot shown in *Figure 8—figure supplement 4C*.

**Figure supplement 5.** Biolayer interferometry analysis of the interaction between PbpC$_{1-13}$ and BacA.

**Figure supplement 6.** Mapping of the PbpC-binding site of BacA by hydrogen-deuterium-exchange (HDX) analysis.

(*Figure 8—figure supplement 4*). Region C1 thus appears to be necessary and sufficient for the recruitment of PbpC to the cell pole-associated BacA assembly, but dispensable for its role in stalk biogenesis.

To verify the ability of region C1 to directly associate with BacA, we generated a synthetic peptide comprising the N-terminal 13 amino acids of PbpC (PbpC$_{1-13}$) and analyzed its interaction with purified BacA polymers using biolayer interferometry. When biosensors carrying the immobilized peptide were titrated with increasing concentrations of BacA, we observed specific binding with an apparent equilibrium dissociation constant ($K_D$) of 4.9 µM (*Figure 8B* and *Figure 8—figure supplement 5*). A similar assay with the polymerization-deficient F130R variant, by contrast, only yielded residual binding with very fast dissociation rates (*Figure 8C*). These results confirm the direct interaction of BacA with region C1 of PbpC. Moreover, they indicate that BacA polymerization is a prerequisite for efficient and stable PbpC binding, likely because the high local accumulation of BacA within polymers leads to an increase in the avidity for its interaction partner.

Having identified a direct association of BacA with the N-terminal peptide of PbpC, we set out to map the PbpC-binding site of BacA using hydrogen-deuterium-exchange (HDX) mass spectrometry, a method that allows the detection of local shifts in the accessibility of backbone amide hydrogens induced by conformational changes and/or ligand binding (*Konermann et al., 2011*). When analyzed alone, the HDX profile of BacA showed very fast HDX in the N- and C-terminal regions flanking the bactofilin domain, which indicates a high degree of disorder and thus corroborates the predicted domain structure of BacA (*Figure 8—figure supplement 6*). Upon incubation with the PbpC peptide, several peptides in the C-terminal half of the central bactofilin domain exhibited a small, yet significant decrease in the HDX rate (*Figure 8C*), with the most prominent changes observed in a region close to the C-terminal polymerization interface (*Figure 8D*). Although additional studies are required to precisely define the interface, this finding suggests that inter-subunit interactions in the bactofilin polymer could potentially induce slight conformational changes in BacA that promote PbpC binding.

To clarify whether the interaction with PbpC contributes to BacA assembly, we compared the localization patterns of BacA-Venus in wild-type and Δ*pbpC* cells. The behavior of the fusion protein was unchanged in the mutant background, suggesting that PbpC does not have a major role in BacA polymerization or membrane binding (*Figure 8—figure supplement 1B*). Nevertheless, it was conceivable that PbpC binding could contribute, at least to some extent, to the association of BacA polymers with the cytoplasmic membrane, thereby complementing the activity of the MTS. To test this hypothesis, we set out to investigate the effect of PbpC on the assembly of an MTS-free BacA-mVenus variant (Δ2–8). Our initial studies showed that the mutant protein had a diffuse localization when analyzed in cells producing PbpC at native levels (*Figure 1B*). However, it was possible that, in this setting, the affinity between the two proteins or the number of PbpC molecules available for the interaction are not be high enough to compensate for the loss of the endogenous membrane-targeting sequence. We, therefore, performed localization studies in cells that overproduced a C-terminally truncated variant of PbpC in which the periplasmic region was replaced by the fluorescent protein mCherry (PbpC$_{1-132}$-mCherry) (*Kühn et al., 2010*), thereby accumulating elevated levels of the PbpC membrane anchor. Control experiments showed that the truncated protein colocalized with wild-type BacA-mVenus and stabilized the polar bactofilin cluster under conditions of BacA-mVenus overproduction (*Figure 9* and *Figure 9—figure supplement 1*). Importantly, the increased availability of the PbpC membrane anchor also fully restored polar localization for the MTS-free fusion protein (Δ2–8), whereas it could not reverse the localization defect of the polymerization-deficient F130R variant (*Figure 9*). These results support the notion that membrane binding and polymerization are cooperative processes that are both required for efficient bactofilin assembly. Moreover, they suggest that membrane-bound client proteins can contribute to the recruitment of bactofilins to the cytoplasmic membrane, although the importance of this effect may vary between systems.

## Discussion

Bactofilins are widespread among bacteria and mediate the spatial organization of diverse cellular processes. Although the polymers they form are typically associated with the inner face of the cytoplasmic membrane, the vast majority of bactofilin homologs lack transmembrane helices, suggesting the existence of an alternative membrane-targeting mechanism. Consistent with this notion, the membrane-binding activity of the bactofilin *Th*Bac from *T. thermophilus* has recently been localized

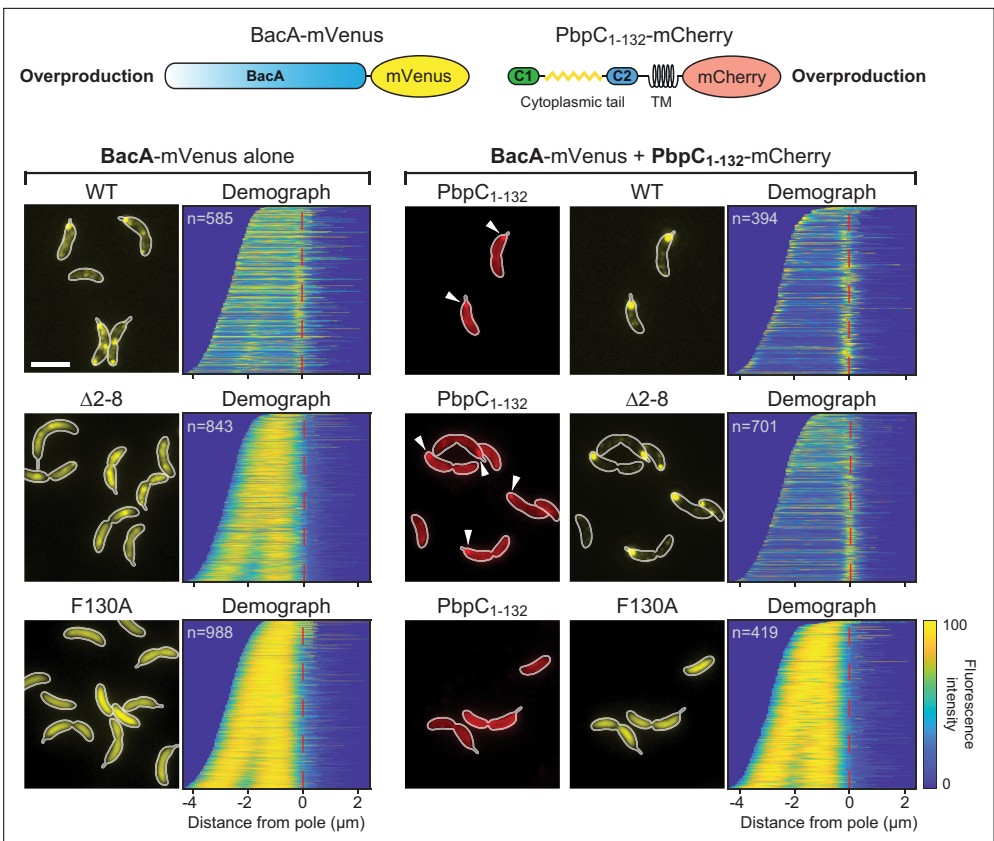

**Figure 9.** Contribution of PbpC to BacA membrane association. *C. crescentus ΔbacAB ΔpbpC* cells producing the indicated BacA-mVenus variants (WT, Δ2–8, F130R) under the control of a xylose-inducible promoter and PbpC₁₋₁₃₂-mCherry under the control of a vanillate-inducible promoter (strains MAB575, MAB576, and MAB577) were grown in the presence of xylose (left) or both xylose and vanillate (right) prior to microscopic analysis. The images show representative fluorescence micrographs, with the cell outlines indicated in white. Arrowheads indicate polar PbpC₁₋₁₃₂-mCherry foci. Demographs summarizing the single-cell BacA-mVenus fluorescence profiles obtained from random subpopulations of cells are provided next to the respective fluorescence images. The number of cells analyzed is shown in the top left-hand corner of each graph. The schematics on top illustrate the protein constructs used for the analysis (not to scale). Scale bar: 3 μm.

The online version of this article includes the following source data and figure supplement(s) for figure 9:

**Figure supplement 1.** Levels and stability the fluorescent fusion proteins used in colocalization studies.

**Figure supplement 1—source data 1.** PDF file containing annotated original images of the Western blots shown in *Figure 9—figure supplement 1A and B*.

**Figure supplement 1—source data 2.** Original images of the Western blots shown in *Figure 9—figure supplement 1A and B*.

to the unstructured N-terminal tail of the protein, and a sequence alignment based on several well-characterized bactofilin homologs suggested the existence of a conserved N-terminal motif in this protein region (*Deng et al., 2019*). Our work considerably extends these initial findings by determining the precise composition of the N-terminal MTS, unraveling its mode of action and clarifying its conservation among bacteria. Moreover, we reveal a mutual dependence of membrane-binding and bactofilin polymerization and provide first insights into mode of interaction between the bactofilin scaffold and membrane-associated client proteins.

Mutational analysis of *C. crescentus* BacA identified residues F2 and K4/K7 as key components of the BacA MTS (*Figure 1B*). This finding is explained by the results of our MD analysis, which show that the insertion of F2 into the hydrophobic core of the lipid bilayer (*Figure 5*) is critical to ensure the proper positioning of the MTS in the membrane, thereby promoting electrostatic interactions of the N-terminal NH₃⁺ group with the negatively charged lipid headgroups of phosphatidylglycerol.

The positively charged ε-amino groups of K4 and K7, by contrast, directly interact with these lipid headgroups, making the largest contribution to the overall interaction energy (*Figure 6*). Notably, the N-terminal methionine residue of proteins is usually removed after translation when followed by an amino acid with a short side chain (*Hirel et al., 1989*). In the case of BacA, however, the bulky phenyl-alanine residue at position 2 presumably inhibits this process, so that residue M1 may be a genuine, conserved part of the MTS. Although M1 interacts most extensively with the lipid bilayer (*Figure 6*), its absence has only a moderate effect on the membrane-binding activity of BacA in vitro (*Figure 3*). This observation supports the idea that most of its interaction energy is contributed by the N-terminal $NH_3^+$ group, whereas the insertion of the M1 side chain into the hydrophobic core of the membrane only serves to tune the positioning of the N-terminus at the lipid-water interface. Consistent with the results of the MD simulations, the loss of either F2 or K4/K7 drastically impairs the membrane-binding activity of BacA (*Figures 2C and 3*). Moreover, it prevents the assembly of BacA into distinct higher-order complexes (*Figure 1B*), suggesting a link between BacA membrane association and polymerization. Indeed, a soluble BacA variant lacking the native MTS (Δ2–8) regains the ability to form polymeric complexes when fused to a heterologous MTS from *E. coli* MreB (*Figure 4C*), indicating that membrane binding is a prerequisite for efficient BacA polymerization. Conversely, polymerization appears to promote membrane binding, because the disruption of the BacA polymerization interface (F130R) strongly increases the fraction of soluble protein (*Figure 4*). Comparable results were obtained when a suitable membrane anchor was provided in trans (*Figure 9*). Together, these findings point to a cooperative binding mechanism, whereby (i) membrane association leads to an increase in the local concentration of BacA that facilitates its self-assembly into polymers, and (ii) polymerization closely juxtaposes multiple MTS-containing N-terminal tails, thereby increasing the avidity of BacA for the membrane and shifting the equilibrium to the membrane-bound state. This process may be particularly relevant at the low bactofilin levels (~200 BacA molecules per cell) found in *C. crescentus* cells (*Kühn et al., 2010*).

Interestingly, substitutions in the MTS that did not completely abolish BacA assembly often led to the formation of multiple and/or mislocalized complexes (*Figure 1—figure supplement 4*). The mechanism determining the polar localization of BacA is still unclear. However, it is conceivable that bactofilin polymers are intrinsically curved and thus preferentially associate with membrane regions of positive or negative Gaussian curvature (*Kühn et al., 2010*; *Pöhl et al., 2024*). Alternatively, the MTS may preferentially bind certain lipid species that are enriched in the polar regions of the cell, with its modification leading to a change in the lipid specificity that interferes with this recruitment pathway. Exchanges at less critical positions of the MTS may affect these recruitment mechanisms by reducing the membrane-binding affinity and thus, indirectly, also the dimensions of BacA polymers, or by altering their lipid specificity.

To assess whether the mode of membrane association established for BacA is more widely conserved, we conducted a comprehensive comparison of the N-terminal regions of all bactofilin homologs known to date. This analysis revealed that amino acid sequences similar to the MTS of BacA are highly prevalent in a large number of bactofilins from the phyla Pseudomonadota, Bacteroidota, and Bacillota, strongly suggesting that MTS-mediated membrane binding is a common theme among this group of proteins (*Figure 7*). Notably, however, most prevalent N-terminal motifs identified for the bactofilin homologs of Spirochaetota and Thermodesulfobacteriota are fundamentally different from those of the other phyla investigated (*Figure 7* and *Figure 7—figure supplement 2*), suggesting that they exhibit in a different mode of membrane binding or interacts with factors other than the cytoplasmic membrane.

Notably, a small subset of bacterial bactofilin homologs contains an N-terminal transmembrane domain, which mediates their stable insertion into the cytoplasmic membrane. While found in a diversity of bacterial phyla, these proteins are particularly abundant in the Pseudomonadota, especially in their gammaproteobacterial and alphaproteobacterial lineages (*Figure 7—figure supplement 1*). One of them, the bactofilin CcmA of *P. mirabilis*, has been previously investigated (*Hay et al., 1999*). It was shown to undergo proteolytic processing and exist in two forms, a full-length form that is an integral membrane protein and a shorter form that lacks the N-terminal transmembrane segments. Intriguingly, the shorter form has an N-terminal sequence (MFSRKTE) that corresponds to the predicted consensus sequence of the MTS in Pseudomonadota and, indeed, behaves like a typical peripheral membrane protein. The specific functional properties conferred by these two

distinct modes of membrane association are still unclear. However, it is tempting to speculate that the assembly of MTS-containing bactofilins may be dynamically regulated by cell cycle-dependent changes in membrane curvature or composition, whereas homologs with transmembrane domains may form more static polymeric assemblies. Consistent with this idea, *C. crescentus* BacA only assembles at the old cell pole once its cell envelope bulges outward to initiate stalk formation, thereby establishing a region of positive Gaussian curvature, although its cytoplasmic levels remain constant throughout the cell cycle (*Kühn et al., 2010*). Similarly, the BacA homolog of the stalked budding bacterium *H. neptunium* consistently assembles in regions of positive Gaussian curvature, first localizing to the stalked pole and then, once the terminal segment of the stalk starts to expand into a bud, to the bud neck (*Pöhl et al., 2024*).

Instead of using an MTS or a transmembrane domain, some bactofilins may also interact with the membrane indirectly by binding to other membrane-associated proteins, as recently suggested for the bactofilin homolog CcmA from *H. pylori* (*Sichel et al., 2022*). Our results show that the interaction with PbpC contributes, at least to some extent, to the membrane association of *C. crescentus* BacA. We did not observe any obvious localization defects in a Δ*pbpC* mutant (*Figure 8—figure supplement 1A*), indicating that PbpC binding is not critical for proper BacA assembly and positioning at native accumulation levels. However, when produced in excess, the PbpC membrane anchor was able to fully compensate for the absence of the N-terminal MTS, providing sufficient links with the cytoplasmic membrane to enable the assembly of polar BacA clusters. Thus, client protein binding is an important factor to consider when analyzing the assembly and localization dynamics of bactofilins in vivo.

The determinants mediating the interaction of membrane-associated bactofilin scaffolds with their client proteins are still poorly understood. For *M. xanthus* BacP, the binding site for its soluble interactor PadC was shown to be located in the exceptionally long unstructured C-terminal tail, which contains a conserved KKKVVVKK motif potentially involved in the interaction (*Anand et al., 2020*; *Lin et al., 2017*). However, most bactofilin homologs have much shorter C-terminal tails, which are typically poorly conserved in sequence and, at least in certain cases, dispensable for function in vivo (*Sichel et al., 2022*), suggesting a direct interaction of client proteins with the bactofilin core domain. Our results indeed indicate that the cytoplasmic tail of PbpC associates with the C-terminal region of the bactofilin domain, close to the polymerization interface. Importantly, BacA assembly appears to be critical for efficient PbpC binding in vitro (*Figure 8C*), suggesting that inter-subunit contacts could potentially change the conformation of BacA, such as to increase its affinity for its client protein. In addition, BacA polymers may have a higher avidity for PbpC than monomers. Polymerization thus acts as a trigger for both membrane association and the recruitment of client proteins, ensuring the tight regulation of bactofilin function.

Collectively, our findings provide important new insights into the function of bactofilins and the nucleotide-independent dynamics of their assembly. It will be interesting to analyze how cell cycle- or stress-induced changes in membrane composition affect the membrane-binding and polymerization behavior of MTS-containing bactofilin homologs. Moreover, it will be informative to shed light on the mode of membrane binding for bactofilins with conserved N-terminal sequences that are clearly distinct from the global consensus motif.

# Materials and methods

## Key resources table

| Reagent type (species) or resource | Designation | Source or reference | Identifiers | Additional information |
|---|---|---|---|---|
| Gene (*Caulobacter crescentus*) | *bacA* | GenBank | ACL95414.1 | |
| Strain, strain background (*Caulobacter crescentus*) | CB15N (aka NA1000) | *Evinger and Agabian, 1977* | ATCC 19089 | *C. crescentus* wild-type strain |
| Genetic reagent (*Caulobacter crescentus*) | JK5 | *Kühn et al., 2010* | | CB15N Δ*bacAB* |
| Genetic reagent (*Caulobacter crescentus*) | MT304 | *Kühn et al., 2010* | | CB15N Δ*pbpC* |

*Continued on next page*

*Continued*

| Reagent type (species) or resource | Designation | Source or reference | Identifiers | Additional information |
|---|---|---|---|---|
| Genetic reagent (*Caulobacter crescentus*) | CB15N derivatives | This paper | | See *Supplementary file 5* |
| Strain, strain background (*Escherichia coli*) | Rosetta(DE3)pLysS | Merck Millipore, Germany | Cat. #: 70956 | F⁻ *ompT hsdSB*(rB⁻ mB⁻) *gal dcm* (DE3) pLysSRARE (Cam^R) |
| Strain, strain background (*Escherichia coli*) | TOP10 | Invitrogen, Germany | Cat. #: C404003 | F⁻ *mcrA* Δ(*mrr-hsdRMS-mcrBC*) Φ80*lacZ*ΔM15 Δ*lacX74 recA1 araD139* Δ(*ara-leu*)7697 *galU galK rpsL* (Str^R) *endA1 nupG* |
| Recombinant DNA reagent | pMT813 | *Kühn et al., 2010* | | pNPTS138 derivative used to generate an in-frame deletion in *bacA*, Kan^R |
| Recombinant DNA reagent | pMT815 | *Kühn et al., 2010* | | pNPTS138 derivative used to generate an in-frame deletion in *bacB*, Kan^R |
| Recombinant DNA reagent | pTB146 | *Bendezú et al., 2009* | | Plasmid for overexpression of protein with N-terminal His₆-SUMO fusion, Amp^R |
| Recombinant DNA reagent | pTB146 derivatives | This paper | | See *Supplementary file 6* |
| Recombinant DNA reagent | pVGFPC-4 | *Thanbichler et al., 2007* | | Integrative plasmid for creating a C-terminal fusion to GFP under the control of P*van*, Gent^R |
| Recombinant DNA reagent | pVGFPC-4 derivatives | This paper | | See *Supplementary file 6* |
| Recombinant DNA reagent | pXCFPC-4 | *Thanbichler et al., 2007* | | Integrative plasmid for creating a C-terminal fusion to CFP under the control of P*xyl*, Gent^R |
| Recombinant DNA reagent | pXmNeonGreenC-4 | This paper | | Integrative plasmid for creating a C-terminal fusion to mNenoGreen under the control of P*xyl*, Gent^R |
| Recombinant DNA reagent | pXmNeonGreenC-4 derivatives | This paper | | See *Supplementary file 6* |
| Recombinant DNA reagent | pXVENC-2 | *Thanbichler et al., 2007* | | Integrative plasmid for creating a C-terminal fusion to Venus under the control of P*xyl*, Kan^R |
| Recombinant DNA reagent | pXmVENC-2 | This paper | | Integrative plasmid for creating a C-terminal fusion to mVenus under the control of P*xyl*, Kan^R |
| Recombinant DNA reagent | pXmVENC-2 derivatives | This paper | | See *Supplementary file 6* |
| Recombinant DNA reagent | pXVENN-1 | *Thanbichler et al., 2007* | | Integrative plasmid for creating an N-terminal fusion to Venus under the control of P*xyl*, Strep/Spec^R |
| Recombinant DNA reagent | pXmVENN-1 | This paper | | Integrative plasmid for creating an N-terminal fusion to mVenus under the control of P*xyl*, Strep/Spec^R |
| Recombinant DNA reagent | pXmVENN-1 derivatives | This paper | | See *Supplementary file 6* |
| Sequence-based reagent | DNA oligonucleotides | This paper | | See *Supplementary file 7* |
| Antibody | anti-FlgH (Antiserum) | *Mohr et al., 1996* | | 1:10,000 |
| Antibody | anti-GFP (Rabbit polyclonal) | Sigma-Aldrich, Germany | Cat. #: G1544, RRID:AB_439690 | 1:10,000 |
| Antibody | anti-MipZ (Antiserum) | *Thanbichler and Shapiro, 2006* | | 1:10,000 |
| Antibody | anti-mNeonGreen (Mouse monoclonal) | Chromotek, Germany | Cat. #: 32f6, RRID:AB_2827566 | 1:1,000 |
| Antibody | anti-RFP (Mouse monoclonal) | MBL Life Science, Germany | Cat. #: M155-3, RRID:AB_1278880 | 1:10,000 |
| Peptide, recombinant protein | PbpC₁₋₁₃ | GenScript, USA | custom-synthesized | Biotin-Ahx-MNDWTLPPYKFDD |
| Chemical compound, drug | ampicillin | Carl Roth, Germany | Cat. #: K029.3 | |
| Chemical compound, drug | chloramphenicol | Carl Roth, Germany | Cat. #: 3886.3 | |
| Chemical compound, drug | gentamicin | Carl Roth, Germany | Cat. #: 0233.3 | |
| Chemical compound, drug | kanamycin | Carl Roth, Germany | Cat. #: T832.3 | |

*Continued on next page*

*Continued*

| Reagent type (species) or resource | Designation | Source or reference | Identifiers | Additional information |
|---|---|---|---|---|
| Chemical compound, drug | streptomycin | Carl Roth, Germany | Cat. #: 0236.3 | |
| Chemical compound, drug | D-xylose | Carl Roth, Germany | Cat. #: 5537.2 | |
| Chemical compound, drug | isopropyl-ß-D-thiogalacto-pyranoside (IPTG) | Carl Roth, Germany | Cat. #: CN08.2 | |
| Chemical compound, drug | 1-palmitoyl-2-oleoyl-sn-glycero-3-phospho-(1'-rac-glycerol) (16:0-18:1 PG) | Avanti Polar Lipids, USA | Cat. #: 840457 C | |
| Software, algorithm | Adobe Illustrator CS6 | Adobe Systems, USA | RRID:SCR_010279 | https://www.adobe.com/products/illustrator.html |
| Software, algorithm | AmphipaSeeK | *Sapay et al., 2006* | | https://npsa-pbil.ibcp.fr/cgi-bin/npsa_automat.pl?page=/NPSA/npsa_amphipaseek.html |
| Software, algorithm | BacStalk | *Hartmann et al., 2020* | | https://drescherlab.org/data/bacstalk/ |
| Software, algorithm | BLAST | *Altschul et al., 1990* | | https://blast.ncbi.nlm.nih.gov/Blast.cgi |
| Software, algorithm | Fiji (2.14.0/1.54 f) | *Schindelin et al., 2012* | RRID:SCR_002285 | https://imagej.net/software/fiji/ |
| Software, algorithm | MEME | *Bailey and Elkan, 1994* | | https://meme-suite.org/meme/tools/meme |
| Software, algorithm | HMMER (3.3.2) | *Eddy, 2011* | | http://hmmer.org/ |
| Software, algorithm | Jalview (version 2) | *Waterhouse et al., 2009* | | https://www.jalview.org/ |
| Software, algorithm | MUSCLE (3.8.31) | *Edgar, 2004* | | https://www.drive5.com/muscle/ |
| Software, algorithm | Oufti | *Paintdakhi et al., 2016* | | https://oufti.org/ |
| Software, algorithm | Pfam | *Paysan-Lafosse et al., 2023* | | https://www.ebi.ac.uk/interpro |
| Software, algorithm | SMTracker (2.0) | *Oviedo-Bocanegra et al., 2021* | | https://sourceforge.net/projects/singlemoleculetracker/ |
| Software, algorithm | u-track (2.2.0) | *Jaqaman et al., 2008* | | https://github.com/DanuserLab/u-track |

## Media and growth conditions

*C. crescentus* CB15N and its derivatives were grown aerobically at 28°C in PYE-rich medium (*Poindexter, 1964*), unless indicated otherwise. To induce stalk elongation, cells were cultivated for 24 hr in M2G$^{-P}$ medium (*Kühn et al., 2010*) prior to analysis. When required, media were supplemented with antibiotics at the following concentrations (µg ml$^{-1}$ in liquid/solid media): kanamycin (5/25), streptomycin (5/5), or gentamicin (0.5/5). *E. coli* strains are grown aerobically at 37°C in LB medium containing antibiotics at the following concentrations (µg ml$^{-1}$ in liquid/solid media): ampicillin (200/200), chloramphenicol (20/30), gentamicin (0.5/5), kanamycin (30/50). To induce gene expression, media were supplemented with 0.005% or 0.03% D-xylose, 0.5 mM sodium vanillate, or 1 mM IPTG, when appropriate.

## Construction of plasmids and strains

The bacterial strains and plasmids used in this work are described in *Supplementary files 5 and 6*. The oligonucleotides used for their construction are listed in *Supplementary file 7*. All plasmids were verified by DNA sequencing. *C. crescentus* was transformed by electroporation (*Ely, 1991*). Non-replicating plasmids were integrated into the chromosome by single homologous recombination at the *xylX* locus (*Thanbichler et al., 2007*). Proper chromosomal integration was verified by colony PCR.

## Live-cell imaging

To prepare samples for microscopy, overnight cultures were diluted to an OD$_{600}$ of 0.1 and cultivated for 1 hr prior to the addition of 0.005% D-xylose. After a further 1 hr (BacA variants) or 2 hr

(PbpC variants) of incubation, the cultures were diluted tenfold, and samples (1.5 µl) of the suspensions were spotted on 1% agarose pads prepared with double-deionized water. Images were taken with an Axio Observer.Z1 microscope (Zeiss, Germany) equipped with a Plan Apochromat 100 x/1.45 Oil DIC, a Plan Apochromat 100 x/1.4 Oil Ph3 M27 objective and a pco.edge 4.2 sCMOS camera (PCO, Germany). An X-Cite 120PC metal halide light source (EXFO, Canada) and appropriate filter cubes (ET-CFP, ET-YFP or ET-TexasRed; Chroma, USA) were used for fluorescence detection. Images were recorded with VisiView 3.3.0.6 (Visitron Systems, Germany) and processed with Fiji 2.14.0/1.54 f (*Schindelin et al., 2012*) and Adobe Illustrator CS6 (Adobe Systems, USA). The subcellular distribution of fluorescence signals was analyzed with BacStalk (*Hartmann et al., 2020*).

## Transmission electron microscopy

BacA or its mutant derivatives (2–3 mg/ml) were dialyzed overnight against TEM buffer (50 mM Tris-HCl (pH 8.0), 200 mM NaCl, 0.1 mM EDTA, 5% glycerol). Subsequently, 5 µl samples of the solutions were applied to glow-discharged carbon-coated grids (G2400C; Plano Wetzlar, Germany). After a brief incubation period, the grids were blotted with filter paper, washed with a droplet of double-distilled water and then treated for ~1 min with 2% (w/v) uranyl acetate before being blotted dry. Micrographs were taken with a JEM-2100 transmission electron microscope (JEOL, Japan) at an acceleration voltage of 120 kV. Images were captured with a 2k × 2k fast scan CCD camera F214 (TVIPS, Germany). Fiji 2.14.0/1.54 f (*Schindelin et al., 2012*) was used for data analysis.

## Single-particle tracking

For single-particle tracking, cells were cultivated in M2G minimal media at 28°C. In the early exponential growth phase, the expression of genes placed under the control of the *xylX* ($P_{xyl}$) promoter was induced by the addition of 0.005% D-xylose. After another 3 h of incubation, cells were spotted on coverslips (25 mm diameter; Menzel Gläser, Germany) and covered with 1% agarose pads prepared with double-distilled water. All coverslips were cleaned before use by sonication in 2% (v/v) Hellmanex II solution (Hellma, Germany) for 30 min at 37°C, followed by rinsing in distilled water and a second round of sonication in double-distilled water. Images were taken at 20 ms intervals by slimfield microscopy (*Plank et al., 2009*), using an Olympus IX-71 microscope equipped with a UAPON 100 x/ NA 1.49 TIRF objective, a back-illuminated electron-multiplying charge-coupled device (EMCCD) iXon Ultra camera (Andor Solis, USA) in stream acquisition mode, and a LuxX 457–100 (457 nm, 100 mW) light-emitting diode laser (Omicron-Laserage Laserprodukte GmbH, Germany) as an excitation light source. The laser beam was focused onto the back focal plane and operated during image acquisition with up to 2 mW (60 W cm$^{-2}$ at the image plane). Andor Solis 4.21 software was used for camera control and stream acquisition. Each single-particle tracking analysis was preceded by the acquisition of a phase contrast image. Subsequently, ~500 frames were acquired to bleach most fluorescent proteins in the cell and thus reach the single-molecule level. Subsequently, remaining and newly synthesized molecules were tracked over ~2500 frames. Prior to analysis, the frames recorded before reaching the single-molecule level were removed from the streams using photobleaching curves as a reference, and the proper pixel size (100 nm) and time increment were adjusted in the imaging metadata using Fiji (*Schindelin et al., 2012*). Subsequently, cell meshes were determined using Oufti (*Paintdakhi et al., 2016*) and single particles were tracked with u-track 2.2.0 (*Jaqaman et al., 2008*). After the removal of all trajectories that were shorter than five steps, the diffusional behavior of the tracked particles was analyzed using SMTracker 2.0 (*Oviedo-Bocanegra et al., 2021*).

## Cell fractionation analysis

Overnight cultures were diluted to an OD$_{600}$ of 0.1 and grown for 2 hr prior to the addition of D-xylose to a final concentration of 0.03%. The cells were further cultivated for 1 hr (*bacA-mVenus* alleles) or 2 hr (*creS-mNeonGreen*) to allow the expression of the genes of interest. Subsequently, cells from 5 ml culture were harvested by centrifugation, washed with 0.2 M Tris-HCl (pH 8.0), and stored at –80°C. For further processing, the pelleted cells were resuspended in a buffer containing 60 mM Tris-HCl (pH 8.0), 0.2 M sucrose, 0.2 M EDTA, 100 µg ml$^{-1}$ phenylmethylsulfonyl fluoride (PMSF) and 10 µg ml$^{-1}$ DNase I and lysed by sonication at 30% amplitude for 7.5 min with alternating 10 s on and off phases (Model 120 Sonic Dismembrator; Fisher Scientific, USA). The lysates were centrifuged for 10 min at 4,000×g (4°C) to remove incompletely lysed cells, followed by ultracentrifugation for 1 hr at 133,000 ×

g (4°C) to separate the soluble and membrane-containing insoluble fractions. The supernatants were carefully removed, and the pellets were resuspended in an equal volume of 0.2 M Tris-HCl (pH 8.0). Both fractions were then subjected to immunoblot analysis with anti-FlgH, anti-MipZ and anti-GFP, or anti-mNeonGreen antibodies.

## Immunoblot analysis

Immunoblot analysis was performed as described previously (*Thanbichler and Shapiro, 2006*). Proteins were detected with a polyclonal anti-GFP antibody (Sigma, Germany; Cat. #: G1544; RRID:AB_439690), a monoclonal anti-mNeonGreen antibody (Chromotek, Germany; Cat. #: 32f6; RRID:AB_2827566), anti-MipZ antiserum (*Thanbichler and Shapiro, 2006*), anti-FlgH antiserum (*Mohr et al., 1996*), or a monoclonal anti-RFP antibody (MBL Life Science, Germany; Cat. #: M155-3) at dilutions of 1:10,000, 1:1000, 1:10,000, 1:10,000, and 1:10,000, respectively. Goat anti-rabbit immunoglobulin G conjugated with horseradish peroxidase (Perkin Elmer, USA) or goat anti-mouse immunoglobulin G conjugated with horseradish peroxidase (Sigma, Germany) were used as secondary antibodies. Immunocomplexes were detected with the Western Lightning Plus-ECL chemiluminescence reagent (Perkin Elmer, USA). The signals were recorded with a ChemiDoc MP imaging system (BioRad, Germany) and analyzed using Image Lab software (BioRad, Germany).

## Protein purification

To overproduce His$_6$-SUMO-BacA or mutant variants thereof, *E. coli* Rosetta(DE3)pLysS cells transformed with the appropriate plasmids were grown overnight at 37°C, diluted 100-fold into 3 l of LB medium containing antibiotics and cultivated to an OD$_{600}$ of ~0.6. Subsequently, the media were supplemented with IPTG to a final concentration of 1 mM, and the cells were incubated for another 4 hr before they were harvested by centrifugation and stored at –80°C until further use. To purify the fusion proteins, the pelleted cells were resuspended in buffer B (50 mM Tris/HCl pH 8.0, 300 mM NaCl, 0.1 mM EDTA, 5% glycerol, and 20 mM imidazole) supplemented with 10 µg/ml DNase I and 100 µg/ml PMSF and lysed by three passages through a French press at 16,000 psi. The crude cell extract was cleared by centrifugation for 30 min at 30,000×g (4°C), and loaded onto a 5 ml HisTrap HP affinity column (GE Healthcare, USA) equilibrated with buffer B with an ÄKTA Pure FPLC system (Cytiva). After washing of the column with 50 ml of buffer B, protein was eluted with a 50 ml linear gradient of imidazole (20–250 mM imidazole in buffer B). Fractions that were highly enriched in the protein of interest were supplemented with 200 µg Ulp1-His$_6$ protease (*Marblestone et al., 2006*) and dialyzed overnight at 4°C against buffer C (50 mM Tris-HCl pH 8.0, 300 mM NaCl, 0.1 mM EDTA, 5% glycerol, 1 mM DTT) to cleave off the His$_6$-SUMO tag and remove the imidazole. Subsequently, the protein solution was again applied to a 5 ml HisTrap HP affinity column to separate the untagged BacA protein from His$_6$-SUMO and Ulp1-His$_6$. The flow-through fractions were collected and analyzed by SDS-PAGE. Fractions containing the desired protein in high amount and purity were dialyzed against buffer D (50 mM Tris-HCl pH 8.0, 200 mM NaCl, 0.1 mM EDTA, 5% glycerol) (BacA$_{\Delta2-8}$) or buffer E (50 mM MOPS-NaOH pH 7.0, 200 mM NaCl, 0.1 mM EDTA, 5% glycerol) (all other BacA variants). The protein solutions were then concentrated using Amicon Ultra Centrifugal Filter (10 kDa MWCO) ultrafiltration devices (Millipore, Germany) and stored in small aliquots at –80°C.

## Size-exclusion chromatography

BacA or its F130R variant were diluted to a concentration of 1.5 mg/ml, applied to a Superdex 200 10/300 GL size-exclusion column (Cytiva, Germany) and eluted at a flow rate of 0.3 ml/min using an ÄKTA Pure FPLC system (Cytiva). Protein was detected by photometry at a wavelength of 280 nm.

## Preparation of liposomes

Liposomes were generated from 1-palmitoyl-2-oleoyl-sn-glycero-3-phospho-(1'-rac-glycerol) (16:0-18:1 PG) (10 mg/mL in chloroform; Avanti Polar Lipids, USA). Chloroform was evaporated in a rotatory evaporator, resulting in a lipid film that was left to dry overnight. Subsequently, the lipids were resuspended in liposome buffer (50 mM MOPS-NaOH pH 7.0, 200 mM NaCl), and the mixture was incubated at room temperature for 1 hr with periodic vigorous agitation. To prepare small unilamellar vesicles (SUVs) with an average diameter of 100 nm, the final lipid solution (20 mg ml$^{-1}$) was extruded

at least ten times through a Mini Extruder (Avanti Polar Lipids, USA) equipped with polycarbonate membranes of 0.1 μm pore size, until the solution was clear.

## Co-sedimentation assay

Protein (20 μM) was incubated for 20 min at room temperature with or without liposomes (0.4 mg ml$^{-1}$) in binding buffer (50 mM MOPS-NaOH pH 7.0, 300 mM NaCl) in a total reaction volume of 100 μl. The mixtures were then centrifuged for 20 min at 100,000×g (20°C) in a TLA-55 rotor (Beckman Coulter, Germany). After transfer of the supernatant to a reaction tube and resuspension of the pellet in 100 μl of liposome buffer (50 mM MOPS-NaOH pH 7.0, 200 mM NaCl), samples of the two fractions were mixed with SDS-PAGE sample buffer, heated at 95°C for 10 min and loaded onto a 15% SDS-polyacrylamide gel. After electrophoresis, protein was stained with Coomassie Brilliant Blue R-250. The gels were imaged in a ChemiDoc MP imaging system (BioRad, Germany) using Image Lab software (BioRad, Germany), and the intensity of protein bands was quantified using Fiji 2.14.0/1.54 f (*Schindelin et al., 2012*).

## Bio-layer interferometry

Bio-layer interferometry experiments were conducted using a BLItz system equipped with Octet High Precision Streptavidin 2.0 (SAX2) Biosensors (Sartorius, Germany). In the initial step, an N-terminally biotinylated PbpC$_{1-13}$ peptide (Biotin-Ahx-MNDWTLPPYKFDD; GenScript, USA) was immobilized on the sensor. After the establishment of a stable baseline, association reactions were monitored with BacA at various concentrations or with different BacA variants of the same concentration. At the end of each binding step, the sensor was transferred into an analyte-free buffer to measure the dissociation kinetics. The extent of non-specific binding was assessed by monitoring the interaction of the analyte with unmodified sensors. All analyses were performed in BLItz binding buffer (50 mM MOPS-NaOH pH 7.0, 100 mM NaCl, 1 mM EDTA, 5% glycerol, 10 μM BSA, 0.01% Tween).

## Hydrogen-deuterium exchange mass spectrometry

Samples were prepared using a two-arm robotic autosampler (LEAP Technologies, Denmark). 7.5 μl of BacA (25 μM) or a mixture of BacA (25 μM) and PbpC$_{aa1-13}$ (100 μM) were mixed with 67.5 μl of D$_2$O-containing buffer (20 mM HEPES-NaOH pH 8.0, 300 mM NaCl) to start the exchange reaction. After 10, 100, 1000, and 10,000 s of incubation at 25 °C, 55 μl samples were taken from the reaction and mixed with an equal volume of quench buffer (400 mM KH$_2$PO$_4$/H$_3$PO$_4$, 2 M guanidine-HCl, pH 2.2) kept at 1 °C. 95 μl of the resulting mixture were immediately injected into an ACQUITY UPLC M-class system with HDX technology (Waters, USA) (*Wales et al., 2008*). Undeuterated samples of BacA and a mixture of BacA and PbpC$_{1-13}$ were prepared similarly by 10-fold dilution into H$_2$O-containing buffer. Proteins were digested online on an Enzymate BEH Pepsin column (300 Å, 5 μm, 2.1 mm × 30 mm; Waters, USA) at 12°C with a constant flow (100 μl min$^{-1}$) of 0.1% (v/v) formic acid in water, and the resulting peptic peptides were collected on a trap column (2 mm ×2 cm) that was filled with POROS 20 R2 material (Thermo Fisher Scientific, USA) and kept at 0.5°C. After 3 min, the trap column was placed in line with an ACQUITY UPLC BEH C18 1.7 μm 1.0×100 mm column (Waters, USA), and the peptides were eluted at 0.5°C using a gradient of 0.1% (v/v) formic acid in water (A) and 0.1% (v/v) formic acid in acetonitrile (B) at a flow rate of 30 μl/min generated as follows: 0–7 min/95–65 % A, 7–8 min/65–15 % A, 8–10 min/15 % A, 10–11 min/5 % A, 11–16 min/95 % A. Peptides were ionized with an electrospray ionization source operated at 250°C capillary temperature and a spray voltage of 3.0 kV. Mass spectra were acquired over a range of 50–2000 m/z on a G2-Si HDMS mass spectrometer with ion mobility separation (Waters, USA) in Enhanced High Definition MS (HDMS$^E$) or High Definition MS (HDMS) mode for undeuterated and deuterated samples, respectively. A [Glu1]-Fibrinopeptide B standard (Waters, USA) was employed for lock mass correction. After each run, the pepsin column was washed three times with 80 μl of 4% (v/v) acetonitrile and 0.5 M guanidine hydrochloride, and blanks were performed between each sample. All measurements were carried out in triplicate.

Peptides from the non-deuterated samples (acquired with HDMS$^E$) were identified with the Protein-Lynx Global SERVER (PLGS, Waters, USA), employing low energy, elevated energy, and intensity thresholds of 300, 100, and 1000 counts, respectively. Peptides were matched using a database containing the amino acid sequences of the proteins of interest, pepsin and their reversed sequences. The search parameters were as follows: peptide tolerance = automatic; fragment tolerance = automatic; min

fragment ion matches per peptide = 1; min fragment ion matches per protein = 7; min peptide matches per protein = 3; maximum hits to return = 20; maximum protein mass = 250,000; primary digest reagent = non-specific; missed cleavages = 0; false discovery rate = 100. Deuterium incorporation was quantified with DynamX 3.0 (Waters, USA), using peptides that fulfilled the following criteria: minimum intensity = 5000 counts; maximum length = 40 amino acids; minimum number of products = 2; maximum mass error = 25 ppm; retention time tolerance = 0.5 min. After automated data processing with DynamX, all spectra were manually inspected and, if necessary, peptides were omitted (e.g. in case of a low signal-to-noise ratio or the presence of overlapping peptides).

## Bioinformatic analysis

Nucleotide and protein sequences were obtained from the National Center for Biotechnology Information (NCBI) or UniProt (*Bateman et al., 2023*) databases, respectively. Sequences were compared and analyzed using the blastn, blastp, or PSI-blast algorithm as implemented on the NCBI website. The presence of putative amphipathic helices was assessed using AMPHIPASEEK (*Sapay et al., 2006*). Multiple sequence alignments were generated with MUSCLE v3.8.31 (*Edgar, 2004*) and viewed and edited with Jalview v2 (*Waterhouse et al., 2009*).

To identify conserved motifs in the N-terminal regions of bactofilins, all known bactofilin homologs were retrieved from the UniProt database (*Bateman et al., 2023*). Subsequently, a Python-based preprocessing procedure was used to exclude non-bacterial entries, remove sequences with predicted transmembrane helices, and eliminate sequences from species with unclear phylogeny. In addition, highly similar sequences (with over 90% similarity) from the same species were filtered out to correct biases due to over-sequencing. The N-terminal region of each protein was determined using hmmscan (HMMER 3.3.2) (*Eddy, 2011*) with the hidden Markov profile constructed from the seed alignment of the bactofilin domain downloaded from the Pfam database (*Paysan-Lafosse et al., 2023*). Subsequently, sequences with N-terminal regions shorter than 8 amino acids were eliminated. The resulting set of sequences was categorized by phylum, and MEME (*Bailey and Elkan, 1994*) was used to detect the 10 most probable motifs. The analysis employed the following settings: classical objective function, with zero or one occurrence per sequence, motif width ranging from 5 to 50 amino acids, and a $0^{th}$-order background Markov model.

## Molecular dynamics simulations

All molecular dynamics (MD) simulations were carried out with Gromacs (version 2021.1) (*Abraham et al., 2015*). The simulation systems were set up and prepared for MD simulation with the Charmm-GUI (*Jo et al., 2008*; *Jo et al., 2009*; *Lee et al., 2016*; *Wu et al., 2014*), using the Charmm36m force field (*Huang et al., 2017*) for the peptide and lipids together with the Charmm-specific TIP3P water model. The lipid composition of the simulated bilayer (*Supplementary file 2*) was chosen such that it mimics as closely as possible the *C. crescentus* membrane (*Chow and Schmidt, 1974*; *De Siervo and Homola, 1980*). Symmetric bilayers with 128 lipids in each monolayer were constructed, that is, the same number and type of lipids was present in the two leaflets. At the beginning of the simulations, a single bactofilin 10-mer peptide (either the wild-type sequence MFSKQAKSNN, or the K4S/K7S or F2Y variants) were modeled as extended conformations and placed in the bulk water phase, located at least 3 nm away from the lipid headgroups. The net charge of the simulation box was neutralized with 150 mM KCl. After energy minimization (with steepest descent), MD simulations were carried out with periodic boundary conditions in the NpT ensemble at a constant temperature of 310 K, maintained via the Bussi velocity-rescaling thermostat (*Bussi et al., 2007*), and constant 1 bar pressure, maintained by semi-isotropic pressure coupling of the lateral (x,y) and normal (z) dimensions of the simulation box to a weak coupling barostat (*Berendsen et al., 1984*). Short-range Coulomb and Lennard-Jones 6,12 interactions were described with a buffered pair list (*Páll and Hess, 2013*) with potentials smoothly shifted to zero at a 1.2 nm distance, with forces switched to zero between 1.0 and 1.2 nm. The long-range electrostatic interactions were described with the particle mesh Ewald (PME) method with 0.12 nm grid spacing (*Darden et al., 1993*). The LINCS and SETTLE algorithms were used to constrain all protein bonds involving H-atoms and to keep the water molecules rigid, respectively, allowing to integrate the equations of motion with 2 fs time steps using the leap-frog integrator. The final production MD simulations were 500 ns long.

## Data analysis

Data were analyzed in Excel 2019 (Microsoft) or Python 3.10.12 and mainly visualized using the Python Matplotlib v3.6.2 (*Hunter, 2007*) and Seaborn v0.12.2 (*Waskom, 2021*) libraries. The plots obtained were edited with Adobe Illustrator CS6 (Adobe Systems) to generate the final figures.

## Statistics and reproducibility

All experiments were performed at least twice independently with similar results. No data were excluded from the analyses. To quantify imaging data, multiple images were analyzed per condition. The analyses included all cells in the images or, in the case of high cell densities, all cells in a square portion of the images. The selection of the images and fields of cells analyzed was performed randomly. Mean values and standard deviations were calculated using Microsoft Excel 2019.

## Availability of biological material

The plasmids and strains used in this study are available from the corresponding author upon request.

## Acknowledgements

We thank Julia Rausch for excellent technical assistance.

## Additional information

### Funding

| Funder | Grant reference number | Author |
| --- | --- | --- |
| University of Marburg | Core funding | Gert Bange<br>Martin Thanbichler |
| Max-Planck-Gesellschaft | Max Planck Fellowship | Gert Bange<br>Martin Thanbichler |
| Deutsche Forschungsgemeinschaft | 450420164 - TH 885/3-1 | Martin Thanbichler |
| Deutsche Forschungsgemeinschaft | EXC 2033 - 390677874 - RESOLV | Lars V Schäfer |
| Deutsche Forschungsgemeinschaft | 324652314 | Gert Bange |
| Deutsche Forschungsgemeinschaft | 260989694 | Gert Bange |
| Marie Sklodowska-Curie Actions | 10.3030/801459 | Saumyak Mukherjee |
| Max-Planck-Gesellschaft | Postdoctoral research grant from the Microcosm Earth Center | Rogelio Hernandez-Tamayo |

The funders had no role in study design, data collection and interpretation, or the decision to submit the work for publication. Open access funding provided by Max Planck Society.

### Author contributions

Ying Liu, Conceived the study. Constructed bacterial strains. Performed cell biological and biochemical studies. Conducted the bioinformatic analyses. Wrote the paper, with input from all other authors; Rajani Karmakar, Performed the MD simulations and analyzed the data obtained; Maria Billini, Constructed strains. Performed cell biological and biochemical studies; Wieland Steinchen, Performed the HDX mass spectrometry analysis; Saumyak Mukherjee, Supervised the MD simulations and analyzed the data obtained. Acquired funding; Rogelio Hernandez-Tamayo, Performed the single-particle tracking analysis. Acquired funding; Thomas Heimerl, Conducted the transmission electron microscopy analysis; Gert Bange, Supervised the HDX analysis. Acquired funding; Lars V Schäfer,

Supervised the MD simulations and analyzed the data obtained. Acquired funding. Contributed to the writing of the manuscript; Martin Thanbichler, Conceived the study. Supervised the cell biological and biochemical analyses and analyzed the data obtained. Acquired funding. Wrote the manuscript, with input from all other authors

### Author ORCIDs
Rogelio Hernandez-Tamayo ⓘ https://orcid.org/0000-0003-4666-3929
Martin Thanbichler ⓘ https://orcid.org/0000-0002-1303-1442

Reviewer #2 (Public review): https://doi.org/10.7554/eLife.100749.3.sa1
Author response https://doi.org/10.7554/eLife.100749.3.sa2

---

# Additional files

### Supplementary files
Supplementary file 1. Diffusion constants of different BacA-mVenus variants. The table shows the number of cells and tracks analyzed in the single-particle tracking studies as well as the diffusion coefficients obtained for each of the proteins investigated.

Supplementary file 2. Composition of the lipid bilayer in the MD simulations. The table shows the type, charge, fatty acid composition, percentage share, and number count of the lipids in each of the two leaflets that constitute the lipid bilayer in the molecular dynamics (MD) simulations.

Supplementary file 3. Representative snapshot from an MD simulation visualizing the interaction of the wild-type $BacA_{1-10}$ peptide with a model lipid bilayer. The file gives the structural coordinates of the snapshot shown in *Figure 5B*.

Supplementary file 4. Detailed description of the HDX data. The spreadsheets give a summary of the conditions used for the HDX analysis and a full list of the peptides obtained in the different experiments.

Supplementary file 5. Strains used in this study. The table gives the genotypes, mode of construction and source of all strains used in this study.

Supplementary file 6. Plasmids used in this study. The table provides descriptions of all the plasmids used in this study, including details of their construction or source.

Supplementary file 7. Oligonucleotides used in this study. The table shows the sequences of all synthetic oligonucleotides used in this study.

MDAR checklist

Source data 1. Source data underlying graphs. This file contains source data for *Figures 3 and 4B*, *Figure 4—figure supplement 1C*, *Figure 8—figure supplement 4*.

### Data availability
All data generated in this study are included in the manuscript, the supplementary material or the source data files.

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
