## [Editor Report · eLife Assessment]

This **valuable** study advances our understanding of how bactofilin cytoskeletal proteins associate with cell membranes by identifying and characterizing a conserved membrane-targeting sequence. The evidence is **solid**, with a well-integrated combination of mutagenesis, biophysical analysis, molecular simulations, and bioinformatics supporting the mechanistic model. The work will be of particular interest to microbiologists and structural biologists studying bacterial cytoskeletons and membrane-protein interactions.

---

## [Referee Report · Reviewer #2 (Public review)]

Summary:

The authors of this study investigated the membrane-binding properties of bactofilin A from Caulobacter crescentus, a classic model organism for bacterial cell biology. BacA was the progenitor of a family of cytoskeletal proteins that have been identified as ubiquitous structural components in bacteria, performing a range of cell biological functions. Association with the cell membrane is a frequent property of the bactofilins studied and is thought to be important for functionality. However, almost all bactofilins lack a transmembrane domain. While membrane association has been attributed to the unstructured N-terminus, experimental evidence had yet to be provided. As a result, the mode of membrane association and the underlying molecular mechanics remained elusive.

Liu at al. analyze the membrane binding properties of BacA in detail and scrutinize molecular interactions using in-vivo, in-vitro and in-silico techniques. They show that few N-terminal amino acids are important for membrane association or proper localization and suggest that membrane association promotes polymerization. Bioinformatic analyses revealed conserved lineage-specific N-terminal motifs indicating a conserved role in protein localization. Using HDX analysis they also identify a potential interaction site with PbpC, a morphogenic cell wall synthase implicated in Caulobacter stalk synthesis. Complementary, they pinpoint the bactofilin-interacting region within the PbpC C-terminus, known to interact with bactofilin. They further show that BacA localization is independent of PbpC.

Although the phenotypic effects of an abolished BacA-PbpC interaction are mild, these data significantly advance our understanding of bactofilin membrane binding, polymerization, and function at the molecular level. The major strength of the comprehensive study is the combination of complementary in vivo, in vitro and bioinformatic/simulation approaches, the results of which are consistent.

---

## [Author Response]

The following is the authors’ response to the original reviews

**Public Reviews:**

**Reviewer #1 (Public review):**
Summary:The investigators undertook detailed characterization of a previously proposed membrane targeting sequence (MTS), a short N-terminal peptide, of the bactofilin BacA in Caulobacter crescentus. Using light microscopy, single molecule tracking, liposome binding assays, and molecular dynamics simulations, they provide data to suggest that this sequence indeed does function in membrane targeting and further conclude that membrane targeting is required for polymerization. While the membrane association data are reasonably convincing, there are no direct assays to assess polymerization and some assays used lack proper controls as detailed below. Since the MTS isn't required for bactofilin polymerization in other bacterial homologues, showing that membrane binding facilitates polymerization would be a significant advance for the field.

We agree that additional experiments were required to consolidate our results and conclusions. Please see below for a description of the new data included in the revised version of the manuscript.

Major concerns(1) This work claims that the N-termina MTS domain of BacA is required for polymerization, but they do not provide sufficient evidence that the ∆2-8 mutant or any of the other MTS variants actually do not polymerize (or form higher order structures). Bactofilins are known to form filaments, bundles of filaments, and lattice sheets in vitro and bundles of filaments have been observed in cells. Whether puncta or diffuse labeling represents different polymerized states or filaments vs. monomers has not been established. Microscopy shows mis-localization away from the stalk, but resolution is limited. Further experiments using higher resolution microscopy and TEM of purified protein would prove that the MTS is required for polymerization.

We do not propose that the MTS is directly involved in the polymerization process and state this more clearly now in the Results and Discussion sections of the revised manuscript. To address this point, we performed transmission electron microscopy studies comparing the polymerization behavior of wild-type and mutant BacA variants. The results clearly show that the MTS-free BacA variant (∆2-8) forms polymers that are indistinguishable from those formed by the wild-type protein, when purified from an *E. coli* overproduction strain (new Figure 1–figure supplement 1). This finding is consistent with structural work showing that bactofilin polymerization is exclusively mediated by the conserved bactofilin domain (Deng et al, Nat Microbiol, 2019). However, at native expression levels, BacA only accumulates to ~200 molecules per cell (Kühn et al, EMBO J, 2010). Under these conditions, the MTS-mediated increase in the local concentration of BacA at the membrane surface and, potentially, steric constraints imposed by membrane curvature, may facilitate the polymerization process. This hypothesis has now been stated more clearly in the Results and Discussion sections.

For polymer-forming proteins, defined localized signals are typically interpreted as slow-moving or stationary polymeric complexes. A diffuse localization, by contrast, suggests that a protein exists in a monomeric or, at most, (small) oligomeric state in which it diffuses rapidly within the cell and is thus no longer detected as distinct foci by widefield microscopy. Our single-molecule data show that BacA variants that are no longer able to interact with the membrane (as verified by cell fractionation studies and in vitro liposome binding assays) have a high diffusion rate, similar to that measured for the non-polymerizing and non-membrane-bound F130R variant. These results demonstrate that a defect in membrane binding strongly reduces the ability of BacA to form polymeric assemblies. To support this hypothesis, we have now repeated all single-particle tracking experiments and included mVenus as a freely diffusible reference protein. Our data confirm that the mobilities of the ∆2-8 and F130R variants are similar and approach those of free mVenus, supporting the idea that the deficiency to interact with the membrane prevents the formation of extended polymeric structures (which should show much lower mobilities). To underscore the relevance of membrane binding for BacA assembly, we have now included a new experiment, in which we used the PbpC membrane anchor (PbpC_1-132_-mCherry) to restore the recruitment of the ∆2-8 variant to the membrane (new Figure 9 and Figure 9–figure supplement 1). The results obtained show that the ∆2-8 variant transitions from a diffuse localization to polar foci upon overproduction of PbpC_1-132_-mCherry. The polymerization-impaired F130R variant, by contrast, remains evenly distributed throughout the cytoplasm under all conditions. These findings further support the idea that polymerization and membrane-association are mutually interdependent processes.

(2) Liposome binding data would be strengthened with TEM images to show BacA binding to liposomes. From this experiment, gross polymerization structures of MTS variants could also be characterized.

We do not have the possibility to perform cryo-electron microscopy studies of liposomes bound to BacA. However, the results of the cell fractionation and liposome sedimentation assays clearly support a critical role of the MTS in membrane binding.

(3) The use of the BacA F130R mutant throughout the study to probe the effect of polymerization on membrane binding is concerning as there is no evidence showing that this variant cannot polymerize. Looking through the papers the authors referenced, there was no evidence of an identical mutation in BacA that was shown to be depolymerized or any discussion in this study of how the F130R mutation might to analogous to polymerization-deficient variants in other bactofilins mentioned in these references.

Residue F130 in the C-terminal polymerization interface of BacA is conserved among bactofilin homologs, although its absolute position in the protein sequence may vary, depending on the length of the N-terminal unstructured tail. The papers cited in our manuscript show that an exchange of this conserved phenylalanine residue abolishes polymer formation. Nevertheless, we agree that it is important to verify the polymerization defect of the F130R variant in the system under study. We have now included size-exclusion chromatography data showing that BacA-F130R forms a low-molecular-weight complex, whereas the wild-type protein largely elutes in the exclusion volume, indicating the formation of large, polymeric species (new Figure 1–figure supplement 2). In addition, we performed transmission electron microscopy analyses of BacA-F130R, which verified the absence of larger oligomers (new Figure 1–figure supplement 1).

(4) Microscopy shows that a BacA variant lacking the native MTS regains the ability to form puncta, albeit mis-localized, in the cell when fused to a heterologous MTS from MreB. While this swap suggests a link between puncta formation and membrane binding the relationship between puncta and polymerization has not been established (see comment 1).

We show that a BacA variant lacking the MTS (∆2-8) regains the ability to form membrane-associated foci when fused to the MTS of MreB. By contrast, a similar variant that additionally carries the F130R exchange (preventing its polymerization) shows a diffuse cytoplasmic localization. In addition, we show that the F130R exchange leads to a loss of membrane binding and to a considerable increase in the mobility of the variants carrying the MTS of *E. coli* MreB. As described above, we now provide additional data demonstrating that elevated levels of the PbpC membrane anchor can reinstate polar localization for the ∆2-8 variant, whereas it fails to do so for the polymerization-deficient F130R variant (new Figure 9 and Figure 9–figure supplement 1). Together, these results support the hypothesis that membrane association and polymerization act synergistically to establish localized bactofilin assemblies at the stalked cell pole.

(5) The authors provide no primary data for single molecule tracking. There is no tracking mapped onto microscopy images to show membrane localization or lack of localization in MTS deletion/ variants. A known soluble protein (e.g. unfused mVenus) and a known membrane bound protein would serve as valuable controls to interpret the data presented. It also is unclear why the authors chose to report molecular dynamics as mean squared displacement rather than mean squared displacement per unit time, and the number of localizations is not indicated. Extrapolating from the graph in figure 4 D for example, it looks like WT BacA-mVenus would have a mobility of 0.5 (0.02/0.04) micrometers squared per second which is approaching diffusive behavior. Further justification/details of their analysis method is needed. It's also not clear how one should interpret the finding that several of the double point mutants show higher displacement than deleting the entire MTS. These experiments as they stand don't account for any other cause of molecular behavior change and assume that a decrease in movement is synonymous with membrane binding.

We now provide additional information on the single-particle analysis. A new supplemental figure shows a mapping of single-particle tracks onto the cells in which they were recorded for all proteins analyzed (Figure 2–figure supplement 1). Due to the small size of *C. crescentus*, it is difficult to clearly differentiate between membrane-associated and cytoplasmic protein species. However, overall, slow-diffusing particles tend to be localized to the cell periphery, supporting the idea that membrane-associated particles form larger assemblies (apart from diffusing more slowly due to their membrane association). In addition, we have included a movie that shows the single-particle diffusion dynamics of all proteins in representative cells (Figure 2-video 1). Finally, we have included a table that gives an overview of the number of cells and tracks analyzed for all proteins investigated (Supplementary file 1). Figures 2A and 4D show the mean squared displacement as a function of time, which makes it possible to assess whether the particles observed move by normal, Brownian diffusion (which is the case here). We repeated the entire single-particle tracking analysis to verify the data obtained previously and obtained very similar results. Among the different mutant proteins, only the K4E-K7E variant consistently shows a higher mobility than the MTS-free ∆2-8 variant, with MSD values similar to that of free mVenus. The underlying reason remains unclear. However, we believe that an in-depth analysis of this phenomenon is beyond the scope of this paper. We re-confirmed the integrity of the construct encoding the K4E-K7E variant by DNA sequencing and once again verified the size and stability of the fusion protein by Western blot analysis, excluding artifacts due to errors during cloning and strain construction.

We agree that the single-molecule tracking data alone are certainly not sufficient to draw firm conclusions on the relationship between membrane binding and protein mobility. However, they are consistent with the results of our other in vivo and in vitro analyses, which together indicate a clear correlation between the mobility of BacA and its ability to interact with the membrane and polymerize (processes that promote each other synergistically).

(6) The experiments that map the interaction surface between the N-terminal unstructured region of PbpC and a specific part of the BacA bactofilin domain seem distinct from the main focus of the paper and the data somewhat preliminary. While the PbpC side has been probed by orthogonal approaches (mutation with localization in cells and affinity in vitro), the BacA region side has only been suggested by the deuterium exchange experiment and needs some kind of validation.

The results of the HDX analysis *per se* are not preliminary and clearly show a change in the solvent accessibility of backbone amides in the C-terminal region in the bactofilin domain in the presence of the PbpC_1-13_ peptide. However, we agree that additional experiments are required to precisely map and verify the PbpC binding site. However, as this is not the main focus of the paper, we would like to proceed without conducting further experiments in this area.

We now provide additional data showing that elevated levels of the PbpC membrane anchor are able to recruit the MTS-free BacA variant (∆2-8) to the cytoplasmic membrane and stimulate its assembly at the stalked pole (Figure 9). These results now integrate Figure 8 more effectively into the overall theme of the paper.

**Reviewer #2 (Public review):**
Summary:The authors of this study investigated the membrane-binding properties of bactofilin A from Caulobacter crescentus, a classic model organism for bacterial cell biology. BacA was the progenitor of a family of cytoskeletal proteins that have been identified as ubiquitous structural components in bacteria, performing a range of cell biological functions. Association with the cell membrane is a common property of the bactofilins studied and is thought to be important for functionality. However, almost all bactofilins lack a transmembrane domain. While membrane association has been attributed to the unstructured N-terminus, experimental evidence had yet to be provided. As a result, the mode of membrane association and the underlying molecular mechanics remained elusive.Liu at al. analyze the membrane binding properties of BacA in detail and scrutinize molecular interactions using in-vivo, in-vitro and in-silico techniques. They show that few N-terminal amino acids are important for membrane association or proper localization and suggest that membrane association promotes polymerization. Bioinformatic analyses revealed conserved lineage-specific N-terminal motifs indicating a conserved role in protein localization. Using HDX analysis they also identify a potential interaction site with PbpC, a morphogenic cell wall synthase implicated in Caulobacter stalk synthesis. Complementary, they pinpoint the bactofilin-interacting region within the PbpC C-terminus, known to interact with bactofilin. They further show that BacA localization is independent of PbpC.Strengths:These data significantly advance the understanding of the membrane binding determinants of bactofilins and thus their function at the molecular level. The major strength of the comprehensive study is the combination of complementary in vivo, in vitro and bioinformatic/simulation approaches, the results of which are consistent.

Thank you for this positive feedback.

Weaknesses:The results are limited to protein localization and interaction, as there is no data on phenotypic effects. Therefore, the cell biological significance remains somewhat underrepresented.

We agree that it is interesting to investigate the phenotypic effects caused by the reduced membrane binding activity of BacA variants with defects in the MTS. We have now included phenotypic analyses that shed light on the role of region C1 in the localization of PbpC and its function in stalk elongation under phosphate-limiting conditions (see below).

**Recommendations for the authors:**

**Reviewer #2 (Recommendations for the authors):**
To address the missing estimation of biological relevance, some additional experiments may be carried out.For example, given that BacA localizes PbpC by direct interaction, one might expect an effect on stalk formation if BacA is unable to bind the membrane or to polymerize. The same applies to PbpC variants lacking the C1 region. As the mutant strains are available, these data are not difficult to obtain but would help to compare the effect of the deletions with previous data (e.g. Kühn et al.) even if the differences are small.

We have now analyzed the effect of the removal of region C1 on the ability of mVenus-PbpC to promote stalk elongation in *C. crescentus* under phosphate starvation. Interestingly, our results show that the lack of the BacA-interaction motif impairs the recruitment of the fusion protein to the stalked pole, but it does not interfere with its stimulatory effect on stalk biogenesis. Thus, the polar localization of PbpC does not appear to be critical for its function in localized peptidoglycan synthesis at the stalk base. These results are now shown in Figure 8–Figure supplement 4. The results obtained may be explained by residual transient interactions of mVenus-PbpC with proteins other than BacA at the stalked pole. Notably, PbpC has also been implicated in the attachment of the stalk-specific protein StpX to components of the outer membrane at the stalk base. The polar localization of PbpC may therefore be primarily required to ensure proper StpX localization, consistent with previous work by Hughes et al. (Mol Microbiol, 2013) showing that StpX is partially mislocalized in a strain producing an N-terminally truncated PbpC variant that no longer localizes to the stalk base.

We have also attempted to investigate the ability of the Δ2-8 and F130R variants of BacA-mVenus to promote stalk elongation under phosphate starvation. However, the levels of the WT, Δ2-8 and F130R proteins and their stabilities were dramatically different after prolonged incubation of the cells in phosphate-limited medium, so that it was not possible to draw any firm conclusions from the results obtained (not shown).

In addition, the M23-like endopeptidase LdpA is proposed to be a client protein of BacA (in C. crescentus, Billini et al. 2018, and H. neptunium or R. rubrum, Pöhl et al. 2024). In H. neptunium, it is suggested that the interaction is mediated by a cytoplasmic peptide of LmdC reminiscent of PbpC. This should at least be commented on. It would be interesting to see, if LpdA in C. crescentus is also delocalized and if so, this could identify another client protein of BacA.

We agree that it would be interesting to study the role of BacA in LdpA function. However, we have not yet succeeded in generating a stable fluorescent protein fusion to LdpA, which currently makes it impossible to study the interplay between these two proteins in vivo. The focus of the present paper is on the mode of interaction between bactofilins and the cytoplasmic membrane and on the mutual interdependence of membrane binding and bactofilin polymerization. Given that PbpC is so far the only verified interaction partner of BacA in *C. crescentus*, we would like to limit our analysis to this client protein.

Further comments:L105: analyze  analyzed

Done.

L169: Is there any reason why the MTS of *E. coli* MreB was doubled?

Previous work has shown that two tandem copies of the N-terminal amphiphilic helix of *E. coli* MreB were required to partially target a heterologous fusion partner protein (GFP) to the cytoplasmic membrane of *E. coli* cells (Salje et al, Mol Cell, 2011).

Fig. S3:a) Please decide which tag was used (mNG or mVenus) and adapt the figure or legend accordingly.b) In the legend for panel (C), please describe how the relative amounts were calculated, as the fractions arithmetically cannot add to > 100%. I guess each band was densiometrically rated and independently normalized to the whole-cell signal?

The fluorescent tag used was mNeonGreen, as indicated in the figure. We have now corrected the legend accordingly. Thank you for making us aware of the wrong labeling of the y-axis. We have now corrected the figure and describe the method used to calculate the plotted values in the legend.

Legend of Fig 1b: It is not clear to me, to which part of panel B the somewhat cryptic LY... strain names belong. I suggest putting them either next to the images, to delete them, or at least to unify the layout (compare, e.g. to Fig S7). (I would delete the LY numbers and stay with the genes/mutations throughout. This is just a suggestion).

These names indicate the strains analyzed in panel B, and we have now clarified this in the legend. It is more straightforward to label the images according to the mutations carried by the different strains. Nevertheless, we would like to keep the strain names in the legend, so that the material used for the analysis can be clearly identified.

Fig. 2a: As some of the colors are difficult to distinguish, I suggest sorting the names in the legend within the graph according to the slope of the curves (e.g. K4E K7E (?) on top and WT being at the bottom).

Thank you for this suggestion. We have now rearranged the labels as proposed.

In the legend (L924), correct typo "panel C" to "panel B".

Done.

Fig. 3: In the legend, I suggest deleting the abbreviations "S" and "P" as they do not show up in the image. In line 929, I suggest adding: average "relative" amount... or even more precisely: "average relative signal intensities obtained..."

We have removed the abbreviations and now state that the bars indicate the “average relative signal intensities” obtained for the different fractions.

Fig 4d: same suggestion as for Fig. 2a.

Done.

Fig 8: In the legend (L978), delete 1x "the"

Done.

L258 and Fig. S5: The expression "To account for biases in the coverage of bacterial species" seems somewhat unclear. I suggest rephrasing and adding information from the M+M section here (e.g. from L593, if this is meant).

We now state that this step in the analysis pipeline was performed “To avoid biases arising from the over-representation of certain bacterial species in UniProt”.

I appreciate the outline of the workflow in panel (a) of Fig. S5. It would be even more useful when some more details about the applied criteria for filtering would be provided (e.g. concerning what is meant with "detailed taxonomic information" or "filter out closely related sequences"). Does the latter mean that only one bactofilin sequence per species was used? (As quite many bacteria have more than one but similar bactofilins.)

We removed sequences from species with unclear phylogeny (e.g. candidate species whose precise taxonomic position has not yet been determined). For many pathogenic species, numerous strains have been sequenced. To account for this bias, only one sequence from clusters of highly similar bactofilin sequences (>90% identity) was retained per species. This information has now been included in the diagram. It is true that many bacteria have more than one bactofilin homolog. However, the sequences of these proteins are typically quite different. For instance, the BacA and BacB from *C. crescentus* only share 52% identity. Therefore, our analysis does not systematically eliminate bactofilin paralogs that coexist in the same species.

L281: Although likely, I am not sure if membrane binding has ever been shown for a bactofilin from these phyla. (See also L 380.) Is there an example? Otherwise, membrane binding may not be a property of these bactofilins.

To our knowledge, the ability of bactofilins from these clades to interact with membranes has not been investigated to date. We agree that the absence of an MTS-like motif may indicate that they lack membrane binding activity, and we have now stated this possibility in the Results and Discussion.

L285: See comment above concerning the M23-like peptidase LpdA. Although not yet directly shown for C. crescentus, it seems likely that BacACc does also localize this peptidase in addition to PbpC. I suggest rephrasing, e.g. "known"  "shown"

We now use the word “reported”.

L295 and Fig S8: PbpC is ubiquitous. Which criteria/filters have been applied to select the shown sequences?

*C. crescentus* PbpC is different from *E. coli* Pbp1C. It is characterized by distinctive, conserved N- and C-terminal tails and only found in *C. crescentus* and close relatives. The *C. crescentus* homolog of *E. coli* PbpC is called PbpZ (Yakhnina et al, J Bacteriol, 2013; Strobel et al, J Bacteriol, 2014), whereas *C. crescentus* PbpC is related to *E. coli* PBP1A. We have now added this information to the text to avoid confusion.

L311: may replace "assembly" by "polymerization"

Done.

L320: bactofilin  bactofilin domain?

Yes, this was supposed to read “bactofilin domain”. Thank you for spotting this issue.

L324: The HDX analysis of BacA suggests that the exchange is slowed down in the presence of the PbpC peptide, which is indicative of a physical interaction between these two molecules. To corroborate the claim that BacA polymerization is critical for interaction with the peptide (resp. PbpC), this experiment should be carried out with the polymerization defective BacA version F130R.(Or tone this statement down, e.g. show  suggest.)

“suggest”

L386: undergoes  undergo

Done.

L391-400: This idea is tempting but the suggested mechanism then would be restricted to bactofilins of C. crescentus and close relatives. The bactofilin of Rhodomicrobium, for example, was shown to localize dynamically and not to stick to a positively curved membrane.

In the vast majority of species investigated so far, bactofilins were found to associate with specifically curved membrane regions and to contribute to the establishment of membrane curvature. Unfortu­nately, the sequences of the three co-polymerizing bactofilin paralogs of *R. vannielii* DSM 166 studied by Richter et al (2023) have not been reported and the genome sequence of this strain is not publicly available. However, in related species with three bactofilin paralogs, only one paralog shows an MTS-like N-terminal peptide and another paralog typically contains an unusual cadherin-like domain of unknown function, as also reported for *R. vannielii* DSM 166. Therefore, the mechanism controlling the localization dynamics of bactofilins may be complex in the *Rhodomicrobium* lineage. Nevertheless, at native expression levels, the major bactofilin (BacA) of *R. vannielii* DSM 166 was shown to localize predominantly to the hyphal tips and the (incipient) bud necks, suggesting that regions of distinct membrane curvature could also play a role in its recruitment. We do not claim that all bactofilins recognize positive membrane curvature, which is clearly not the case. It rather appears as though the curvature preference of bactofilins varies depending on their specific function.

L405-406: I agree that localization of BacA has been shown to be independent of PbpC. However, this does not generally preclude an effect on BacA localization by other "client" or interacting proteins. (See also comment above about the putative BacA interactor LpdA). I suggest either to corroborate or to change this statement from "client binding" to "PbpC binding".

Thank you for pointing out the imprecision of this statement. We now conclude that “PbpC binding” is not critical for BacA assembly and positioning.

Suppl. Fig. S11: In the legend, please correct the copy-paste mismatch (...VirB...).

Done.

L482: delete 1x "at"

Done.

L484: may be better "soluble and insoluble fractions"?

We now describe the two fractions as “soluble and membrane-containing insoluble fractions” to make clear to all readers that membrane vesicles are found in the pellet after ultracentrifugation.

L489-490: check spelling immunoglobulin – immuneglobulin

Done.

L500 and 504: º_C  ºC

Done.

Suppl. file X (HDX data): please check the table headline, table should be included in Suppl. file 1

We have now included a headline in this file (now Supplementary file 3).